# Strange semimetal dynamics in SrIrO$_3$

K. Sen[1], D. Fuchs[1], R. Heid [1], K. Kleindienst [1], K. Wolff[1], J. Schmalian[1,2] & M. Le Tacon [1✉]

The interplay of electronic correlations, multi-orbital excitations, and spin-orbit coupling is a fertile ground for new states of matter in quantum materials. Here, we report on a polarized Raman scattering study of semimetallic SrIrO$_3$. The momentum-space selectivity of Raman scattering allows to circumvent the challenge to resolve the dynamics of charges with very different mobilities. The Raman responses of both holes and electrons display an electronic continuum extending far beyond the energies allowed in a regular Fermi liquid. Analyzing this response within a memory function formalism, we extract their frequency dependent scattering rate and mass enhancement, from which we determine their DC-mobilities and electrical resistivities that agree well with transport measurement. We demonstrate that its charge dynamics is well described by a marginal Fermi liquid phenomenology, with a scattering rate close to the Planckian limit. This demonstrates the potential of this approach to investigate the charge dynamics in multi-band systems.

[1] Institut für Quantenmaterialien und -technologien, Karlsruher Institut für Technologie, 76021 Karlsruhe, Germany. [2] Institut für Theorie der Kondensierten Materie, Karlsruher Institut für Technologie, 76131 Karlsruhe, Germany. ✉email: matthieu.letacon@kit.edu

Numerous quantum materials have exotic electronic properties that cannot be accounted for within the canonical framework of the Fermi-liquid theory. They attract increasing attention both because of the profound challenge they pose to our fundamental understanding of electrons in condensed matter[1] and because of their technological potential[2]. Over the past decade, the spin–orbit coupling (SOC), which describes the strength of the interaction between the spin and the orbital motion of a quasiparticle, has been identified as one of the major ingredient for the realization of novel quantum phases of matter[3], encompassing in particular topological or axion insulators, and quantum spin liquids, as well as Dirac or Weyl topological semimetals. The interplay between electronic correlations and semimetal behavior has, for example, been discussed in the context of quantum criticality in Dirac systems[4,5] and topological and non-Fermi-liquid states that emerge near quadratic band touching points[6].

The 5$d$ transition metal oxides are particularly interesting materials in this respect. In the iridate perovskites from the Ruddlesden–Popper series $Sr_{n+1}Ir_nO_{3n+1}$, the crystal field lifts the degeneracy of the 5$d$ levels of the octahedrally coordinated $Ir^{4+}$ ions. Combined with electron–electron correlations ($U \sim$ 2 eV) and large SOC (~0.4 eV), this yields a peculiar type of insulating antiferromagnetic phases at half-filling in $Sr_2IrO_4$ ($n =$ 1) and $Sr_3Ir_2O_7$ ($n = 2$) compounds. This phase is known as the spin–orbital Mott state[7], in which $J_{eff} = 1/2$ pseudo-spins rather than pure spins order magnetically and is widely seen as a novel platform for unconventional superconducting states. In addition, a plethora of fascinating exotic physical phenomena, encompassing pseudogap or Fermi arc states, have been discovered in these systems and remain to be understood[8]. The $n = \infty$ phase of this series, $SrIrO_3$ (SIO), exhibits, on the other hand, a semimetallic and paramagnetic ground state[9–11], and has long been predicted to host Dirac quasiparticles near the Fermi energy $E_F$[12,13], making it a potential realization of a correlated Dirac semimetal[14].

Recent angle-resolved photoemission spectroscopy (ARPES) studies on SIO directly confirmed the semimetallic character of this compound, in which both a heavy hole-like and a lighter electron-like band cross the Fermi level $E_F$[15,16], as predicted by first-principles calculations[12,13,17]. Despite a very steep and quasi-linear dispersion of the electron-like band, the theoretically predicted[12,13] symmetry-protected degeneracy of the Dirac point has been found to be lifted[16]. Although evidences for the proximity of SIO to a magnetic insulating state have been reported[18–20], the sizeable mixing of the $J_{eff} = 1/2$ and 3/2 character of the narrow bands crossing $E_F$ strongly contrasts with the case of $Sr_2IrO_4$.

Significant efforts have been devoted to understand the charge dynamics in this system. However, disentangling the contribution of both types of carriers in such multiband systems is generally a delicate task using conventional transport methods. In the particular case of SIO, since the effective mass of quasi-linearly dispersing electrons is much smaller than the one of the heavier holes[15], the electrons have a higher mobility ($\mu = e\tau_0/m^*$, $\tau_0$: static relaxation time) and dominate Hall effect measurements[11,21,22]. Another study suggests that the electrical conductivity and thermopower are affected by both electron- and hole-like carriers[23]. Both types of charge carriers further contribute to the optical conductivity[11,24,25].

Consequently, and to the best of our knowledge, neither the static nor the dynamical scattering rates and mass enhancements of the electron- and hole-like carriers were determined experimentally in SIO. Investigations of the charge dynamics in this system are further complicated by the fact that, unlike the $n = 1$ and 2 compounds from which single crystals can be grown, the perovskite phase of SIO is metastable and can only be synthesized in polycrystalline or in thin film forms. Subtle structural differences between the polycrystalline materials and the thin films can yield different behavior at low temperatures (electronic structure reconstruction, location of the Dirac node, metal-to-insulator transition, sign of the magnetoresistance, etc.) and generally illustrate the strong sensitivity of the electronic properties of this system to structural details[20,26]. Even for a given substrate, slight changes in the lattice parameters can induce metal-to-insulator transition[27]. For most of the substrates, significant lattice mismatches induce large strain effects, as well as, for example, twin or grain boundaries that considerably affect the electrodynamics of the system. The Hall coefficient, the carrier density, or the resistivity have recently been reported to be strongly substrate and film thickness dependent[18,22,28,29], and also indicate strong strain–relaxation effects. As both first-principles calculations[12,13] and ARPES[15,16] investigations have shown that the hole and electron pockets crossing the Fermi level occupy distinct regions of the reciprocal space, the way towards momentum-selective studies of the charge dynamics[30–32] in SIO via electronic Raman scattering (ERS) is paved.

In this communication, we take full advantage of the polarization selection rules of ERS to independently investigate the intrinsic charge dynamics of the hole- and electron-like carriers in a fully strain-relaxed 50-nm-thick SIO thin film. Exploiting the confocality of our micro-Raman setup, we have been able to extract the electronic response from the thin film and to reveal the existence, for both types of charge carriers, of a flat electronic continuum extending at least up to 1000 cm$^{-1}$ (~125 meV). This is strongly reminiscent of the Raman response of doped high-$T_c$ superconducting cuprates[33–35]. This in turn suggests that such continuum might be a universal feature of correlated electron system on the verge of a Mott transition. The reported electronic continuum is a characteristic feature of the marginal Fermi-liquid (MFL) phenomenology[36] and has been analyzed using a memory function formalism, which allowed us in particular to quantitatively extract charge-carrier-resolved frequency-dependent scattering rates $\Gamma(\omega, T)$ and mass enhancements $m^*/m_b = 1 + \lambda(\omega, T)$, where $m^*$ and $m_b$ are the effective mass renormalized by electron–electron (e–e) interactions and the bare band mass, respectively. For both type of carriers, these quantities amount for a rather broad temperature regime to an inverse quasiparticle time

$$\hbar\tau^{-1} = \Gamma/(1 + \lambda), \qquad (1)$$

which is surprisingly close to the Planckian limit[37,38] $\tau_\hbar^{-1} = k_B T/\hbar$ (even though at lowest $T$ the MFL theory yields $\tau^{-1} \propto T/\log(D/T)$, where $D$ is an appropriate cut-off frequency as discussed in Supplementary Note 2). More generally, this work demonstrates the potential of polarized ERS for the study of scattering rate, mass enhancement, and mobility of charge carriers in semimetals and other multiband systems.

## Results

**Samples.** Epitaxial thin films of $SrIrO_3$ (50 nm) were grown by pulsed laser deposition on orthorhombic ($Pnma$) (101)-oriented $DyScO_3$ (DSO) substrates and characterized, as described in ref. [21]. As shown in Fig. 1a, our untwinned films are (101) oriented, with their $b$-axis parallel to the one of DSO substrate. The tilt pattern of the $IrO_6$ octahedra, $a^-b^+a^-$ in Glazer notation is the same as that of bulk $SrIrO_3$. The choice of DSO substrates, which has the closest lattice parameters to that of bulk SIO, minimizes epitaxial strain effects. Our recent magnetoresistance measurements on similar DSO-grown SIO films with 9 and 50 nm indicate that the electric transport is already strain independent, at least from the thickness of 9 nm onwards[39].

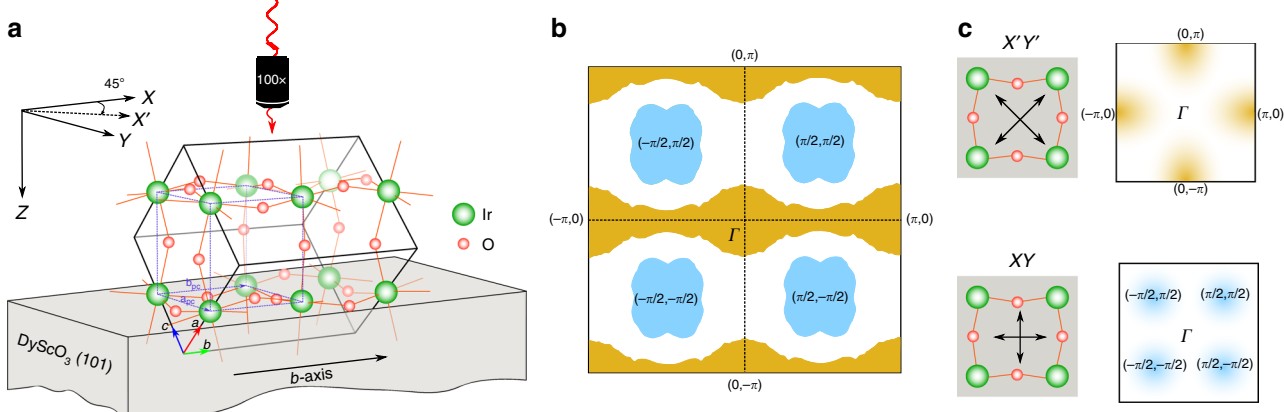

**Fig. 1 Polarization selection rules. a** Sketch illustrating orientation of $SrIrO_3$ crystal unit cell of *Pnma* space group on (101)-$DyScO_3$ substrate. A pseudocubic unit cell of $SrIrO_3$ is designated by blue dashed lines. The *XYZ*-coordinate system concerning the backscattering Raman experiments is marked in black arrows. Incoming laser propagates along *Z*-axis, as shown by a red wavy-arrow through a microscope objective. **b** Calculated Fermi surface is shown in 1-Ir (pseudocubic unit cell) Brillouin zone (BZ) for the respective pseudocubic unit cell. The electron pockets are at $(\pm\pi/2, \pm\pi/2)$, whereas the rest arises from hole-like bands. Note that the hole pocket at the $\Gamma$-point and at the 1-Ir BZ boundary at $(\pm\pi, 0)$ are equivalent. **c** Electronic Raman scattering structure factors are shown on 1-Ir BZ (right panels) for the respective scattering configurations, namely $Z(X'Y')\bar{Z}$ and $Z(XY)\bar{Z}$, as depicted in left panels. The corresponding symmetries are $E_g$ and $T_{2g}$ according to the point group of $O_h$, respectively.

**Fermi surface**. We have calculated the Fermi surface of the (101)-oriented SIO with respect to its pseudocubic unit cell (UC), in which the Ir atoms sit at the corners of the pseudocube using first-principles calculations, as depicted in Fig. 1b. In agreement with previous reports and ARPES studies[15,16], we find multiple electron bands crossing the Fermi level around $(\pm\pi/2, \pm\pi/2)$, and several flat hole bands extending along the $\Gamma$–$X$ directions from the $\Gamma$ and $Y$ points.

**Polarization selection rules**. Confocal Raman scattering experiments were carried out in backscattering geometry. As detailed in the "Method" section (and in the Supplementary Figs. 1 and 2), the confocal geometry allows us to accurately separate the intrinsic Raman response of the SIO thin film from that of the DSO substrate[40,41]. Figure 1c summarizes the Raman polarization selection rules relevant for the present study, which allow to selectively probe the charge dynamics on different sections of the Fermi surface of SIO. The *XYZ*-coordinate system used to label the polarization of the light is oriented along the Ir–Ir bonds of the pseudocubic UC (incident laser propagates in the direction perpendicular to the plane of the film, along the *Z*-axis). The Raman vertex will be accordingly described within the cubic $O_h$ point group. When the incident photon polarization is at 45° to the Ir–O–Ir bonds and the scattered photons are detected in a crossed-polarization configuration (Porto's notation: $Z(X'Y')\bar{Z}$), the corresponding electronic Raman structure factor has the $E_g$ symmetry $(x^2 - y^2)$, which probes the dynamics of charge carriers with momentum along the $(0, \pm\pi)$ and $(\pm\pi, 0)$ directions of the 1-Ir Brillouin zone (BZ), as shown in Fig. 1c. Similarly, the electronic Raman structure factor in $Z(XY)\bar{Z}$ has the symmetry of $T_{2g}$ $(xy)$, which probes the charge carriers whose momentum is located around the $(\pm\pi/2, \pm\pi/2)$ in 1-Ir BZ, as shown in Fig. 1c. Thus, in summary, $X'Y'$ and $XY$ geometries predominantly probe dynamics of hole and electron pockets, respectively.

**Raman responses in $X'Y'$ and $XY$**. Figure 2a, b display the Raman response $\chi''(\omega)$ of the SIO film in $X'Y'$ and $XY$ geometries, respectively, at several temperatures (*T*). In both cases, $\chi''(\omega)$ consists of sharp phonons that are superimposed to a flat continuum extending over the whole range of investigated frequencies. Before discussing the origin of this continuum, it is crucial to check that it does not arise from an artifact related to

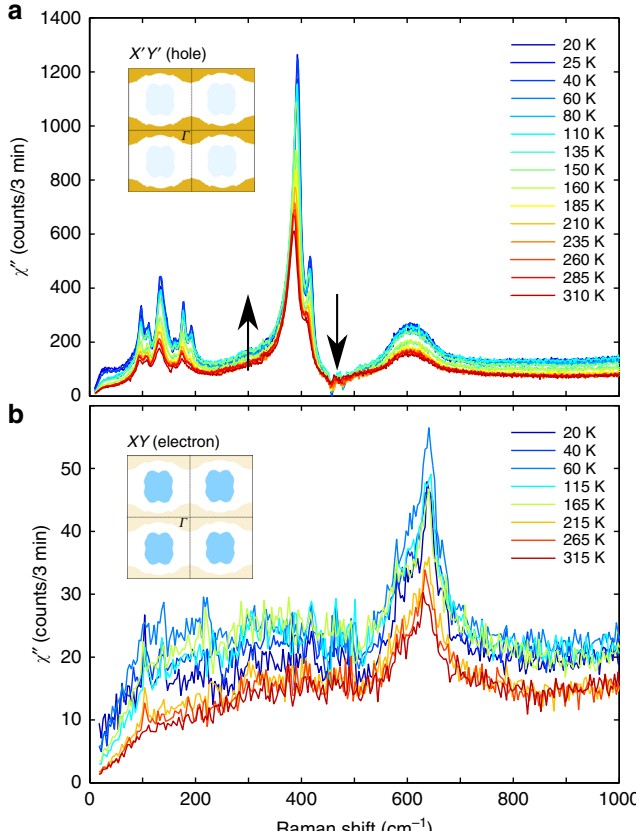

**Fig. 2 Symmetry-dependent Raman spectra. a** Raman spectra ($\chi''(\omega)$) in $X'Y'$ at several temperatures. The arrows indicate spectral weight loss and gain above and below the phonon modes ~400 cm$^{-1}$, respectively. **b** The corresponding plot in $XY$ at a few temperatures.

the subtraction of the substrate's contribution. To do this, we applied the same procedure to a 60-nm-thick film of insulating $CeO_2$ (see Supplementary Fig. 3), in which no continuum in the Raman response is expected. Its absence in the experimental spectrum therefore confirms that the featureless Raman response

observed in SIO is intrinsic. This continuum is strongly reminiscent of the electronic Raman response encountered in high-$T_c$ cuprates[30–35] or Fe-based superconductors[42,43], in which it arises from the creation of particle–hole excitations across the Fermi level ($E_F$)[30]. Our interpretation for SIO is supported by the observation that the continuum is ~5 times bigger in $X'Y'$ than in XY, as qualitatively expected from the larger density of states (DOS) of the flat hole-like bands at $(0, \pm\pi)$ and $(\pm\pi, 0)$ in comparison to the almost linearly dispersing electron-like bands around $(\pm\pi/2, \pm\pi/2)$.

**Phonon contribution.** In order to analyze the $X'Y'$ Raman continuum in details, it is necessary to subtract the other recorded spectral features (Fig. 2a). We observe nine peaks (7 between 90 and 220 cm$^{-1}$, and 2 more ~400 cm$^{-1}$) that sharpen and harden as temperature is lowered and that correspond to Raman active lattice vibrations of SIO. Their detailed analysis will be the subject of a separate study. The broader peak centered around 600 cm$^{-1}$ is only weakly temperature dependent, and its frequency is larger than the calculated cut-off frequency of the phonon spectrum (560 cm$^{-1}$ at the $\Gamma$ point). We therefore tentatively assess this to a double-phonon process (typically two phonons from the high DOS region of dispersion with opposite momenta) and its contribution to the Raman response can be easily subtracted. A close inspection of the two most intense modes ~400 cm$^{-1}$ reveals that the lower energy one is significantly asymmetric, that is, spectral weight gain toward low frequency seemingly occurs at a cost of spectral weight loss at high frequency (highlighted by arrows in Fig. 2a). This lineshape, known as a Fano resonance[44], generally arises from the coupling between a discrete excitation and a continuum. Crucially here, this implies that the electronic continuum cannot be analyzed independently and that its coupling to the phonon should explicitly be taken into account (see "Methods"). The contribution of all the other phonon lines can be modeled using regular Lorentzian lineshapes. The analysis in the XY channel is comparatively simple, since the corresponding spectra have only a broad feature >600 cm$^{-1}$, which we also attribute to double-phonon scattering. It can be easily subtracted out and that does not distort the underlying $e$-continuum scattering background.

**Electronic Raman scattering.** To extract the pure electronic contribution $\chi''_e(\omega, T)$ from the $X'Y'$ data, we follow the approach described in ref. [45] and use a phenomenological model (see Eq. (8) in "Methods") to fit the Raman response $\chi''(\omega, T)$. Importantly, this approach (described in more details in "Methods") requires an analytical form for the $\chi''_e(\omega, T)$ that we express in terms of a memory function $M(\omega, T)$. The method was originally introduced by Götze and Wölfle[46] to calculate the frequency-dependent optical conductivity of metals, and has been generalized to extract dynamical scattering rate $\Gamma(\omega, T)$ and mass enhancement factor $1 + \lambda(\omega, T)$ of charge carriers. It was adapted to analyze ERS in high-$T_c$ cuprates[47] and Fe-based superconductors[43]. In the memory function formalism, the experimentally challenging determination of the ERS intensity in absolute units is not required and has not been attempted here. Note that in the following, only electron–electron scattering contribution to the total scattering rate (in particular in the static limit) are included, and possible contributions of long wavelength acoustical phonons to the total scattering rate have not been considered (see Supplementary Note 5 for a detailed discussion).

The memory function parametrisation of the total electronic Raman response function reads[47]:

$$\chi_e(\omega, T) = \frac{M(\omega, T)}{\hbar\omega + M(\omega, T)}, \quad (2)$$

The real ($\chi'_e$) and imaginary ($\chi''_e$) parts of $\chi_e$ are even and odd functions of the Raman shift $\omega$, respectively. Thus, we can write

$$M(\omega, T) = \hbar\omega\lambda(\omega, T) + i\Gamma(\omega, T), \quad (3)$$

where $\lambda(\omega, T)$ and $\Gamma(\omega, T)$ are real even functions that are related via a Kramers–Kronig relation.

$\chi''_e(\omega, T)$ follows from Eqs. (2) and (3) as

$$\chi''_e(\omega, T) = \frac{\hbar\omega\Gamma(\omega, T)}{[\hbar\omega(1 + \lambda(\omega, T))]^2 + [\Gamma(\omega, T)]^2}. \quad (4)$$

In analogy to the optical conductivity, we can introduce the quasiparticle scattering time $\tau$ of Eq. (1), which includes the mass renormalization, contained in the real part of the memory function, that is, in $\lambda$, see, for example, ref. [48].

To proceed, we analyze specific forms of the scattering rate $\Gamma(\omega, T)$ and determine $\lambda(\omega, T)$ via Kramers–Kronig transformation. $\Gamma(\omega, T)$ essentially contains two terms, arising from temperature-independent impurity scattering ($\Gamma_{imp}$) and electron–electron scattering, respectively. For a conventional metal (Fermi liquid), this electron–electron scattering rate is given by:

$$\Gamma_{FL}(\omega, T) = \frac{g}{E_F}\left[(\hbar\omega)^2 + (\beta k_B T)^2\right], \quad (5)$$

where the dimensionless coupling constant $g$ characterizes the correlation strength[49]. $\beta$ is a numerical coefficient that determines the relative importance of thermal vs. dynamic excitations for the scattering rate. In the case of a single particle scattering rate holds $\beta = \pi$, while a quantum Boltzmann analysis for the optical conductivity, a two-particle quantity, yields $\beta = 2\pi$[50,51]. The analysis of refs. [52,53] revealed that, in principle, $\pi \leq \beta < \infty$ is possible, depending on the relative strength of elastic and inelastic scattering processes. This was for instance reported for URu$_2$Si$_2$[54] and UPt$_3$[55].

The dynamics of both types of charge carriers inferred from our data cannot be adequately described in terms of this Fermi-liquid expression (see inset of Fig. 3a and Supplementary Note 2). This yields our first important conclusion, namely, charge dynamics in SIO is non-Fermi liquid like. To gain further insights, we therefore modeled the Raman response in terms of a singular quasiparticles scattering rate as $\Gamma(\omega, T) \propto ((\hbar\omega)^2 + (\beta k_B T)^2)^\alpha$, with an exponent $\alpha \neq 1$. Acceptable fits of our data can be achieved with values for $\alpha$ in the range $0.45 \leq \alpha \leq 0.6$ (see Supplementary Note 2). For simplicity we chose $\alpha_{MFL} = 1/2$, which corresponds to the MFL phenomenology[36]. This choice is also motivated by the fact that the observed featureless ERS continuum of SIO is strongly reminiscent to that of the high-$T_c$ cuprates[33–35], in which it is considered as one of the hallmark of the MFL phenomenology[35,36]. Crucially, the dependence of the different fitting parameters on the exact value of $\alpha$ lies well within the corresponding error bars. Choosing $\alpha$ in the vicinity of $\alpha_{MFL}$ will merely change some logarithmic dependencies to power laws with small exponents.

**MFL phenomenology.** The characteristics of the MFL is that the quasiparticle weight in the single particle spectrum vanishes like $1/\log\left(\frac{D}{\hbar\omega}\right)$, that is, the spectrum tends to be weakly incoherent near the Fermi energy[36]. According to refs. [35,36], the scattering rate of the Raman response $\Gamma(\omega, T)$ for an MFL can be written as

$$\Gamma_{MFL}(\omega, T) = g\sqrt{(\hbar\omega)^2 + (\beta k_B T)^2}, \quad (6)$$

where $g$ is again a dimensionless strength of the coupling. Microscopically it is determined by the ratio of the effective e–e correlations, that is, onsite Coulomb repulsion and the

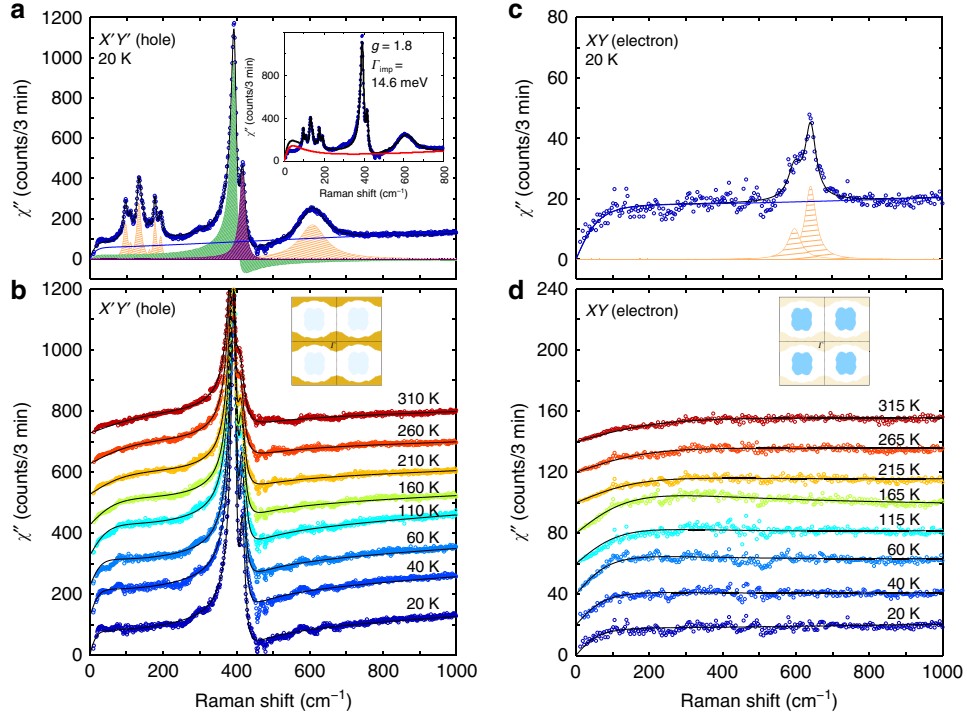

**Fig. 3 Fits with MFL model to extract electronic Raman scattering response. a** Raman spectrum at 20 K in $X'Y'$ (blue circles), with the details of the fit to our phenomenological marginal Fermi-liquid model (solid line in black, see text). The solid line in blue is the best-suited $e$-continuum scattering according to the marginal Fermi-liquid (MFL) model in Eq. (4). Phonon modes are shaded in orange (Lorentzian lineshape) and green or purple (Fano lineshapes). The inset displays a representative attempt to fit these data with Fermi-liquid model. **b** Temperature dependence of the $X'Y'$ Raman spectra, after subtraction of the symmetrical (Lorentzian) phonon lines. The data were vertically shifted for representation. The solid lines are the best-suited $e$-continuum scattering backgrounds according to the MFL model, including the coupling to the phonons ~400 m$^{-1}$. **c** Raman spectra ($\chi''(\omega)$) in $XY$ at 20 K. **d** Temperature dependence of the electronic part or the Raman response in $XY$. The data were vertically shifted for representation. The solid lines are the best-suited $e$-continuum scattering backgrounds according to the MFL model.

bandwidth. $\beta$ plays an analogous role as for the Fermi-liquid rate discussed above.

The total scattering rate is then given by:

$$\Gamma(\omega, T) = \left[\Gamma_{MFL}(\omega, T) + \Gamma_{imp}\right]\phi(\omega/D), \quad (7)$$

where $\phi$ is an appropriate cut-off function with a cut-off frequency of $D$. We have set $\hbar D$ to the bandwidth of each charge carrier. In both cases its value amounts to several hundreds of meV, and thereby only weakly affects $\Gamma(\omega, T)$ in the investigated range of frequencies (see Supplementary Note 2). More details regarding the memory function formalism based on the MFL model are given in the Supplementary Note 2. Representative resulting global fits (including phonon lines and their eventual coupling to the continuum) to $\chi''(\omega)$ between $T = 20$ and 310 K for $X'Y'$ are shown in Fig. 3a and b, respectively. The agreement with the data both at low and high temperatures is excellent and validates the MFL approach to describe the charge dynamics for the hole-like carriers in SIO. We note that the Fano asymmetry parameter for the 391 cm$^{-1}$ mode is essentially temperature independent in the investigated range, indicating no significant change in the electronic structure at the Fermi surface as a function of temperature. This contrasts with the recent optical conductivity report[11] that has been interpreted as a significant reconstruction of the electron pockets near $E_F$. We note however that this was observed on polycrystalline samples, with different structural parameters than films, in which these effects are strongly reduced[27]. Similarly, the data recorded in the $XY$ channel (Fig. 3c, d) can also be very well fitted in this framework.

**Analysis of Raman spectra with the MFL model.** In Fig. 4a, b, we display the result of our analysis, that is, the pure electronic Raman responses $\chi''_e(\omega)$ in $X'Y'$ and $XY$ channels, respectively. The responses in the two channels appear quite distinct at all temperatures. This is particularly striking at energies $\hbar\omega \gg k_B T$, in which we detect an increase of $\chi''_e(\omega)$ in $X'Y'$, which, in contrast, remains essentially constant in $XY$. To better understand the origin of such difference, we take a closer look at the values of the parameters $g$, $\beta$, and $\Gamma_{imp}$ obtained for each Fermi pockets from our fitting procedure.

First of all, we note that the $\Gamma_{imp}$ and $\beta$ parameters are both temperature-independent quantities, but both can take very different values for the different Fermi pockets. From our analysis, they respectively amount to $\Gamma_{imp}^{hole} = 12.6 \pm 2$ meV and $\beta^{hole} = 1.84 \pm 0.3$ (~0.6$\pi$) in $X'Y'$ and $\Gamma_{imp}^{elec} = 30.6 \pm 5$ meV and $\beta^{elec} = 3.23 \pm 0.5$ (~$\pi$) in $XY$.

As shown in Supplementary Fig. 7a, the value of the dimensionless coupling constant $g$ (the ratio of the onsite Coulomb repulsion to the bandwidth) is only weakly temperature dependent. It is at least two times larger in $X'Y'$ ($g \sim 1.2$) than in $XY$ ($g \sim 0.5$), indicating stronger e–e correlations on the hole pockets compared to the electron pockets in SIO. This difference can be qualitatively understood as a consequence of the smaller DOS for electron carriers (1.6 states/eV per UC from our calculation) compared to the hole ones (10.7 states/eV per UC), which naturally reduces interaction effects. The values for the electron pockets are similar to those reported for superconducting cuprates YBa$_2$Cu$_3$O$_7$ ($g = 0.55$) and Bi$_2$Sr$_2$CaCu$_2$O$_8$ ($g = 0.4$)[35]. Thus, it seems that despite much lower onsite Coulomb repulsion

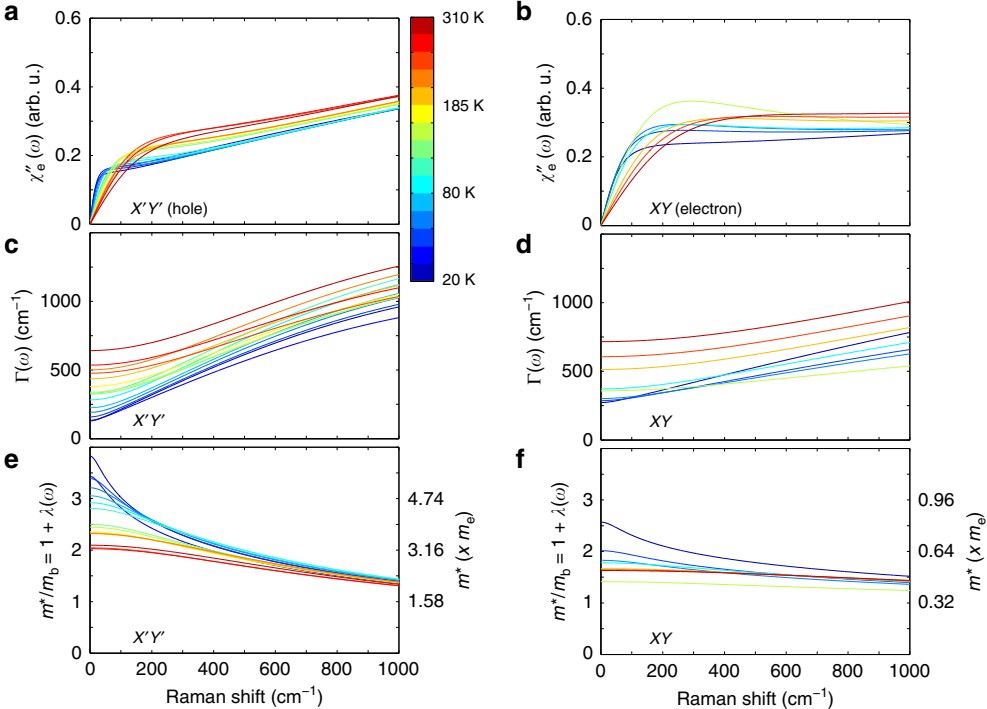

**Fig. 4 Electronic Raman scattering and charge transport parameters from the MFL fits.** Electronic Raman scattering response ($\chi''_e(\omega)$) at several temperatures in **a** $X'Y'$ and **b** $XY$. Dynamical scattering rates ($\Gamma(\omega, T)$) in **c** $X'Y'$ and **d** $XY$. The corresponding dynamical mass enhancement factors ($m^*/m_b(\omega, T)$) in **e**, **f**, respectively. They were obtained from the marginal Fermi-liquid (MFL) fits, as shown in Fig. 3.

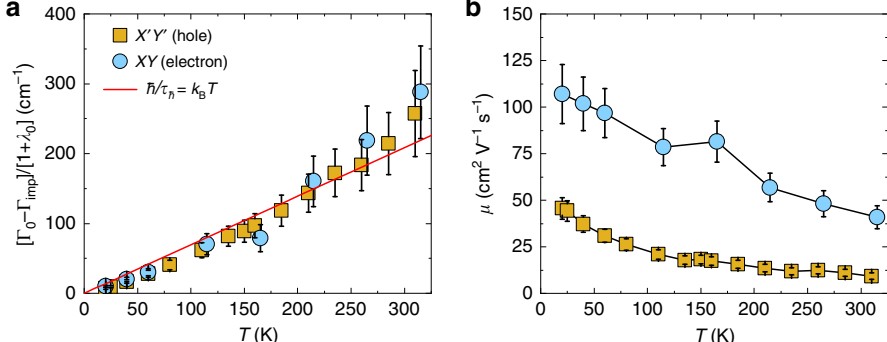

**Fig. 5 Static charge transport properties. a** Inverse inelastic quasiparticle scattering time of conduction holes and electrons as function of temperature, both following the universal Planckian limit ($\hbar\tau_\hbar^{-1} = k_B T$). **b** Calculated DC mobility for both charge carriers as function of temperature. The error bars reflect the maximum proportional error calculated from the errors on individual fitting parameters (see Supplementary Note 2).

the hole pockets in SIO are effectively at least as correlated as the cuprates' charge carriers. This is a consequence of the large spin–orbit interaction, which reduces the electronic bandwidth and lowers the critical Coulomb repulsion[56] required to induce the transition from the current paramagnetic metallic state to the nearby Mott-insulating state[18–20].

This has a direct impact on the frequency dependence of the dynamical scattering rate $\Gamma(\omega, T)$ and mass renormalization $1 + \lambda(\omega, T) = m^*/m_b(\omega, T)$ of the two types of carriers. Indeed, the larger $g$ in $X'Y'$ compared to $XY$ results in a steeper $\Gamma(\omega, T)$ for the holes than for the electrons, as shown in Fig. 4c and d, respectively. A larger $g$ value also enhances the effective mass of the holes more than that of the electrons (see Fig. 4e, f), yielding an effective mass enhancement $m^*_0/m_b = 1 + \lambda_0 = 1 + \lambda(\omega \to 0, T)$ of 3.8 for the former against 2.6 for the latter at low temperatures.

**Transport parameters in the static limit.** To allow direct comparison with the other experiments, we estimate transport parameters such as the resistivity and mobility of each type of charge carrier from $\Gamma(\omega, T)$ and $1 + \lambda(\omega, T)$ in the $X'Y'$ and $XY$ channels in the static limit ($\omega \to 0$), respectively. Under the assumption that the rates for impurity and inelastic scattering are additive, we obtain for both carrier types that the inverse inelastic quasiparticle time $\hbar\tau_{\text{inel.}}^{-1} = (\Gamma_0 - \Gamma_{\text{imp}})/(1 + \lambda_0)$ is very close to the Planckian scattering bound[37,38], $k_B T$, as shown in Fig. 5a. This bound plays an important role in the discussion of transport quantities[48] where it serves as a measure of the degree of correlation strength of a quantum material. The subsequent comparison of $\tau^{-1}$ with transport parameters strongly suggests that the rate obtained by Raman scattering allows for the same conclusion. Our observation that $\tau_{\text{inel.}} \sim \tau_\hbar$ for holes and electrons, characterized by distinct coupling constants $g$, cut-off frequencies $D$,

and coefficients $\beta$, is remarkable. At the lowest temperatures, the logarithmic growth of the MFL mass renormalization yields $\hbar\tau^{-1} \sim \frac{\pi\beta}{2}\frac{k_B T}{\log\frac{D}{\beta T}}$, where holes and electrons reach different quasi-particle rates. In the temperatures range investigated here, however, the MFL mass renormalization remains limited $\lambda(T) < 3$ (see Supplementary Note 2), and the log divergence of $\hbar\tau^{-1}$ is not detectable. At this point, it is unclear whether the observed "universality" of $\tau$ for different momenta is a coincidence or whether there is a deeper underlying principle at work.

The knowledge of the static quasiparticle scattering time $\tau_0 = \hbar(1 + \lambda_0)/\Gamma_0$, together with the assumption that it behaves similar to the one in the electric conductivity, allows for an estimate of the electron mobility $\mu = \frac{e\tau_0}{m_0^*}$ ($e$ is the electronic charge) and of the Drude-like DC resistivity $\rho_0 = m_0^*/(ne^2\tau_0)$ of each type of charge carrier. The former quantity can be directly extracted from our analysis from which we determined the temperature-dependent mass enhancement (with respect to the calculated band masses $m_{b,h} = 1.58m_e$ and $m_{b,e} = 0.32m_e$, see "Methods") and static rates (Fig. 4), while the latter requires the knowledge of the charge-carrier densities, as discussed below. We summarize the static transport parameters extracted from our analysis at room temperature and 20 K in Table 1, and we plot the extracted mobility of the two charge carriers in Fig. 5b. Despite a larger value of the coupling constant $g$ for the holes compared to the electrons, the static relaxation rate $\Gamma_0(T) = g\beta k_B T + \Gamma_{imp}$ is comparable for the two types of carriers (see Supplementary Fig. 7b), given the smaller value of $\beta$ and $\Gamma_{imp}$ for the holes.

The drastically different mobilities of the two types charge carriers naturally account for the electron-like linear and negative Hall resistance ($R_{XY}$ vs. $H$) measured on our sample[21] and previously reported in the literature[15,23]. To detect the field-dependent non-linearities of the Hall resistance arising from the multiband nature of the system, magnetic fields much >15 T would be required (see Supplementary Note 3). This nevertheless implies that the estimate of the charge-carrier density inferred from $R_{XY}$, assuming that the Hall resistance is only caused by electron carriers, is overestimated. The value obtained for this sample at 2 K, $n_e = 3.0 \times 10^{20}$ cm$^{-3}$ is only weakly temperature dependent (it increases by ~6% at room temperature), and is of the same order of magnitude, albeit larger, than the estimate of ref. [23] ($n_e = 1.6 \times 10^{20}$ cm$^{-3}$). Note that the sample of ref. [23] was grown on a different substrate, and that the transport properties have been reported to be substrate dependent[22]. Based on this, and on the assumption that $n_h \sim 1.5 \times n_e$ (as in ref. [23]), we have estimated the resistivity for each charge carrier. The total resistivity can be calculated by assuming that the two channels conduct current as parallel resistors, yielding $\rho_0(20\text{ K}) \sim 0.1$ m$\Omega$ cm and $\rho_0(300\text{ K}) \sim 0.4$ m$\Omega$ cm. Thus, the value we determine here for the room temperature resistivity of our SIO film is reasonably close to that determined using conventional resistivity measurements (see Supplementary Fig. 9), and is consistent with the values of 0.5–2 m$\Omega$ cm reported in the literature[15,18,28,29]. Given, as discussed above, that the Hall measurements tends to

overestimate the electron charge-carrier density, both values for the resistivity extracted from the ERS measurements are slightly underestimated, and the agreement with transport is overall very satisfactory at room temperature. The situation is a bit different at low temperature where our transport measurement indicate an upturn in the resistivity, which is not seen here. This upturn is associated with weak localization effects that occur over the length scale probed in the transport measurement, and to which the present experimental approach, which probes the system over a length scale of ~2 μm (set by the size of our laser spot), is not sensitive. The proposed approach therefore allows us to extract the intrinsic dynamics of the quasiparticles in SIO.

## Conclusion

To summarize, in this work we combined the selection rules of polarized Raman scattering and the high spatial resolution of confocal geometry to investigate independently the dynamics of electron- and hole-like charge carriers in a strain-relaxed film of semimetallic SrIrO$_3$ as thin as 50 nm. We find that neither of them can be described within the framework of the Fermi-liquid theory, and that the electronic Raman response can be well modeled using marginal Fermi-liquid (MFL) phenomenology. Using mass enhancement and the DC scattering rate obtained from this analysis allow us to retrieve the mobility for the two types of charge carriers. The results confirmed the much larger mobility of the electron carriers that generally dominate transport experiments. A next natural step would be to investigate how structural degrees of freedom impact this charge dynamics, in particular as SrIrO$_3$ is driven across metal-to-insulator and/or magnetic transitions, using for instance lattice strain tuning. The proposed approach more generally demonstrates the power of Raman scattering to resolve the dynamics of charge carriers in correlated semimetals and more generally multiband systems.

## Methods

**Polarization-resolved confocal Raman scattering**. Confocal Raman scattering experiments were performed with a Jobin–Yvon LabRAM HR evolution spectrometer in backscattering geometry. We used a He-Ne laser ($\lambda = 632.8$ nm) with a laser power of ≤0.8 mW that was focused on the sample with a ×100 magnification long-working-distance (7.6 mm) objective. The laser spot size was ≈2 μm in diameter. The sample was placed on a motorized stage that translate along the incident beam direction which, in combination with a 50 μm confocal hole at the entrance of the spectrometer, provides the high spatial resolution required to investigate thin films. To do so, we first optimized the signal from the film by recording Raman spectra of the sample at different position across the focal point of the microscope. Reference spectra from the substrate was obtained focusing the laser deep in the substrate, and subtracted from the total response (Supplementary Note 1). Spectra were recorded using a grating of 600 grooves/mm, yielding the spectrometer resolution of 1.8 cm$^{-1}$. Any resonance effect in the spectral background was discarded by measuring with a second laser of $\lambda = 532$ nm. Temperature-dependent measurements were carried out by placing the samples in a He-flow Konti cryostat. All representative spectra were corrected for Bose factors and laser-induced heating of ≤15 K, estimated from systematic study of Raman data with laser power.

**Gobal fit**. To disentangle the electronic contribution from the total (electron + phonon) Raman response $R\chi''(\omega)$ in $X'Y'$ configuration, we fitted the data using the approach proposed by Chen et al.[45]. The Raman response is given by:

$$R\chi''(\omega) = R\left[\chi_e''(\omega) + \sum_{i=1}^{2}\frac{g_i^2}{\Lambda_i(\omega)\left[1+\epsilon_i^2(\omega)\right]}\left[S_i^2(\omega) - 2\epsilon_i(\omega)S_i(\omega)\chi_e''(\omega) - \chi_e''^2(\omega)\right]\right]$$
$$+ R\left[\sum_{j=1}^{8}\frac{C_j w_j}{2\pi}\frac{1}{\left(\omega-\omega_{c,j}\right)^2+\left(\frac{w_j}{2}\right)^2}\right],$$
$$(8)$$

where $R$ is a proportionality constant, which is a function of temperature in the present case, imaginary part ($\chi_e''(\omega)$) of the pure electronic Raman response is entangled with two phonon modes through the Fano terms expressed in the second term of Eq. (8), and the last term accounts for the remaining eight phonon modes that have Lorentzian lineshapes. $\epsilon_i(\omega)$ and $S_i(\omega)$ are defined as $\epsilon_i(\omega) = (\omega^2 - \Omega_i^2)/2\omega_{0,i}\Lambda_i(\omega)$ and $S_i(\omega) = S_{0,i} + \chi_e'(\omega)$, respectively. Here, $\chi_e'(\omega)$ is the real part

---

**Table 1 DC mass enhancement factor ($m^*/m_b$), DC scattering rate $\Gamma_0 = \Gamma(\omega \rightarrow 0)$, static quasiparticle relaxation time ($\tau_0 = \hbar(1 + \lambda_0)/\Gamma_0$), mobility ($\mu$), and resistivity ($\rho_0$) of the conduction holes and electrons around 310 and 20 K.**

| | $m_0^*/m_b = 1 + \lambda_0$ | $\Gamma_0$ (cm$^{-1}$) | $\tau_0$ (×10$^{-15}$ s) | $\mu$ (cm$^2$ V$^{-1}$ s$^{-1}$) | $\rho_0$ (mΩ cm) |
|---|---|---|---|---|---|
| Hole (RT) | 2.1 | 642 | 17.3 | 9.2 | 1.4 |
| Electron (RT) | 1.6 | 716 | 12.1 | 40.7 | 0.5 |
| Hole (20 K) | 3.8 | 130 | 157 | 45.6 | 0.3 |
| Electron (20 K) | 2.6 | 273 | 50 | 107 | 0.2 |

of the electronic Raman response, and was obtained from Eq. (2). Renormalized phonon frequency and linewidth in the presence of electron–phonon coupling $(g_i)$ are defined as $\Omega_i^2 = \omega_{0,i}^2 - 2\omega_{0,i}g_i^2\chi_e'(\omega)$ and $\Lambda_i(\omega) = \Lambda_{0,i} + g_i^2\chi_e''(\omega)$, respectively. $\omega_{0,i}$ and $\Lambda_{0,i}$ are the intrinsic phonon frequency and linewidth, respectively. $S_{0,i}$ combines Raman phononic matrix element $(T_{p,i})$, Raman electronic matrix element $(T_{e,i})$ and $g_i$, and it reads $S_{0,i} = T_{p,i}/(T_{e,i} \cdot g_i)$. $c_j$, $w_j$, and $\omega_{c,j}$ in the last term of Eq. (8) are area, linewidth, and center of the $j$th Lorentzian phonon, respectively. The parameter that determines the Fano asymmetry reads

$$q_i(\omega) = -\frac{S_i(\omega)}{\chi_e''(\omega)}, \qquad (9)$$

in which $1/q_i^2(\omega_{0,i})$ describes the asymmetry in the phonon lineshape.

**DFT calculation**. DFT calculations were performed for SrIrO$_3$ using the mixed-basis pseudopotential method[57] (Meyer, Elsässer, and Fähnle. *FORTRAN90 Program for Mixed-Basis Pseudopotential Calculations for Crystals*, unpublished). This method combines plane waves and local functions in the basis set, which allow an efficient description of more localized components of the valence states. In this study, we used plane waves up to a kinetic energy of 28 Ry, augmented by local functions of $s$, $p$, and $d$ type at the Sr and Ir sites, respectively, and of $s$ and $p$ type at the O sites. Norm-conserving pseudopotentials were constructed following the description of Vanderbilt[58], including the Sr-4$s$, Sr-4$p$, Ir-5$s$, Ir-5$p$, and O-2$s$ semi-core states in the valence space. The local-density approximation in the para-metrization of Perdew and Wang[59] has been employed. Spin–orbit interaction is consistently incorporated in the DFT Hamiltonian by using a spinor formulation and by including spin–orbit components of the pseudopotentials[60]. For the orthorhombic structure of SrIrO$_3$, we took the experimental lattice parameters of the film on the DyScO$_3$ substrate[21]. Atomic positions were relaxed until the remaining forces were below $10^{-3}$ Ry/a.u. For the relaxation, it was sufficient to employ an orthorhombic $8 \times 8 \times 6$ $k$-point mesh for BZ integration in conjunction with a Gaussian smearing of 0.1 eV. Due to the presence of flat hole-like bands close to the Fermi level, band structure and subsequent Fermi surfaces were determined with the tetrahedron method without any broadening, and with a very dense $32 \times 32 \times 24$ $k$-point mesh.

The Fermi surface shown in Fig. 1b represents a top view for a (101) orientation of SrIrO$_3$. The electron pockets at $(\pm\pi/2, \pm\pi/2)$ in 1-Ir BZ are a robust feature associated to steep bands, whereas the hole-like Fermi surface is an open surface arising from very flat bands and is very sensitive to structural details. Rough estimates for Fermi velocities $v_F$ of holes and electrons have been derived from the band dispersion along high-symmetry directions. Effective band masses were estimated using the formula: $m_b = (\hbar q_F)/v_F$, where $q_F$ denotes the wavevector of the Fermi level crossing measured with respect to the pocket centers. This procedure gave average band masses for holes and electrons of $1.58m_e$ and $0.32m_e$, respectively.

## Data availability
The Raman data reported in this study have been deposited at the KIT Open, under the following identification number KITopen-ID: 1000119920. Any further relevant data are available from the authors on request to M.L.T. (matthieu.letacon@kit.edu).

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

## Acknowledgements

We are grateful to I. Paul, Y. Gallais, and R. Eder for valuable discussions. We acknowledge support by the state of Baden-Württemberg through bwHPC.

## Author contributions

K.S. and M.L.T. conceived the project. K.S. acquired and analyzed the Raman scattering data under the supervision of M.L.T. SrIrO$_3$ thin films were prepared and characterized by D.F., K.K., and K.W. R.H. performed first-principles calculations to obtain Fermi surface. J.S. developed the theoretical framework that was used to fit the Raman scattering data. K.S., J.S., and M.L.T. wrote the manuscript with the inputs from all the co-authors.

## Funding

## Competing interests

The authors declare no competing interests.
