## [Peer Review File · Nature Communications]

Editorial Note: Parts of this peer review file have been redacted as indicated to remove third-party material where no permission to publish could be obtained.

Reviewers' comments:

Reviewer #1 (Remarks to the Author):

This manuscript reports Raman measurements in polarized light on the SrIrO₃ thin films that focus on the electronic Raman response. A careful and methodical analysis of the electronic Raman responses using the memory function formalism originally used by Opel et al in 2000 and developed here, leads the authors to identify two kinds of carriers, hole and electron pockets, depending on the polarizations of light used.

They say that the Raman responses of these carriers cannot be interpreted as a Fermi Liquid but rather as a marginal Fermi liquid. From this point of view, they consider that the charge dynamic in SrIrO₃ is similar to that of strongly correlated systems and they found that the static scattering rate versus T follows the Planckian limit observed from transport data in cuprates. By this study the authors highlight the power of Raman spectroscopy for studying the carrier dynamics and in particular, scattering rate, effective mass of charge carriers.

The authors have carried out a very fine experimental study and an analysis of the Raman data using the formalism of the memory function which opens new perspectives for the exploitation of electron Raman spectroscopy.

However, there are several points which concern me and which do not seem clear enough for me to be in favour of an immediate publication.

The first point is the authors' assertion that Raman response cannot be interpreted in the context of a Fermi liquid model.

The authors show curve fits in figures S4 and S5 of the supplementary information without giving the values of the parameters. what are the values of g , D , β , and λ ?

To what extent by varying g and /or λ in the FL model , it would not be possible to better fit the low energy part of the Raman spectra in figures S4 and S5 and so attenuate or even make the hump disappear around 100 cm⁻¹?

Importantly, if one of the main conclusion of the paper is that the dynamics of the carriers in SrIrO₃ thin films cannot be interpreted in terms of FL expression. I expect that at least, the S4 or S5 figure (with the parameters used) to be put in the main text and not in the supplementary information.

In figure 3, which show the different stage of the process for extracting the Raman electronic background in X'Y' configuration, is hard to understand why in panel (c) is plotted the temperature dependence of the electronic XY Raman response and not the one of X'Y'? Can the authors show the electronic XY Raman response and if they can't do it, they have to explain why?

In view of the electronic Raman data, the authors propose that SrIrO₃ is a strongly correlated material as it is the case of the cuprates. If this is actually the case, I would like the authors indicate or suggest which kind of strong electronic interactions occur in SrIrO₃? SrIrO₃ is a non-magnetic material, the correlations likely do not come from antiferromagnetic fluctuations as in cuprates, so, why make the analogy with cuprates? I would like the authors to be clearer on this point.

Finally, I have a couple of questions:

-How the authors know the system they studied is free of constraints between the substrate and the thin film?

-how the authors eliminate the Raman vibration modes of the air molecules at the second focus point in the confocal set-up?

-Why the phonon peaks are smaller in intensity in the XY configuration with respect to the X'Y' configuration?

-The authors mentioned a double phonon around 600 cm⁻¹ but I don't see any single phonon at 300 cm⁻¹. Can the authors explain why?

Reviewer #2 (Remarks to the Author):

This paper presents the study of electronic dynamics in the strongly correlated Dirac semimetallic SrIrO₃. Combined with the inelastic light scattering and ab-initio calculation, authors analyzed the electronic excitation in the scheme of the marginal Fermi liquid phenomenology. They show that the spectral shape can be modelled by using the marginal Fermi liquid theory, but the temperature dependence of scattering rate is not fully understood by the existing model. Although there are some issues to be addressed, the results are interesting. In my opinion, this manuscript may deserve the consideration in Nature Communications, after addressing the following issues.

1) SrIrO₃ is a Dirac semimetal which possesses electrons with Dirac-like dispersion and holes with conventional band dispersion. An important issue of this material is the dynamics of Dirac electrons. Authors clarified that the renormalization is slightly smaller for Dirac electrons than for holes. Is there a special mechanism that the renormalization of Dirac electron is smaller than that of holes?

2) Authors present that the temperature dependence of scattering time appears to be proportional to T in all the temperature region and does not show the "low-T regime of marginal Fermi liquid theory". This is an interesting result, but the reason is not clear. Why is "the low-T regime" missing in this material? The materials nearby the quantum critical point often show such T-linear behavior. Is SrIrO₃ related to the quantum criticality?

3) The ARPES study reveals that there is a hole pockets nearby the Gamma-point in the momentum space, which substantially contributes the DC transport. In this work, authors do not mention the hole pocket nearby Gamma-point since the Raman scattering probes the pockets at the zone boundary. Nevertheless, in the final part of the main text and Figure 4, authors compare the results of Raman scattering with that of DC-transport. The missing information about hole pocket at Gamma-point may be critical for this comparison. It would be better to append the explanation about the comparison.

4) Authors suggest that the electronic structure at the Fermi surface does not significantly change as a function of temperature. However, this statement may not be consistent with the previous research by optical conductivity which reveals a substantial change of electronic structure as a function of temperature. It would be better to append some explanation.

Reviewer #3 (Remarks to the Author):

The manuscript, entitled "Strange semimetal dynamics in SrIrO₃", brings a Raman study of SrIrO₃, a member of iridate perovskites from the Ruddlesden-Popper series. Unlike other members of that series, which are layered, SrIrO₃ is a three-dimensional system, exhibiting (semi)metallic behavior.

The two most important achievements presented in this manuscript are:

- i) Independent measurements of the electron- and hole-like contributions to the Raman response, by probing different sections of the FS.
- ii) A fitting procedure that provides a relatively convincing case for a phenomenological description of the charge dynamics in terms of a scattering rate that is linear in frequency and temperature.

I think that these results are interesting and eventually worth publishing in Nature Communications. However, I have some issues with the consistency of the new findings with the already published results. Furthermore, I have worries regarding the physical interpretation of the experimental data. In the light of these concerns, I would like to draw attention to few essential problems, mostly in relation to the nature of the scattering rate. Specifically:

i) How important are the strain-relaxation effects for the present study? In the part of the manuscript that discusses experimental aspects it is said that fully strain-relaxed 50 nm thick films are measured, suggesting that the electron band structure and the Raman response are not affected by the film thickness. Are there any data to confirm this?

ii) It appears that there is a good agreement in the literature that bulk SrIrO₃ exhibits a metallic resistivity [PRB 89, 214106; PRB 95, 121102(R); J. Appl. Phys. 103, 103706]. In particular, in the recent study [PRB 89, 214106], for low temperatures ($T < 30\text{K}$), it was found that the bulk resistivity follows the typical Fermi-liquid behavior quadratic in temperature. Above these temperatures the resistivity became linear in T , which is typical for metals and for the scattering on thermal phonons. The authors of [PRB 89, 214106] find no indication for an upturn in resistivity below 50 K reported in [J. Appl. Phys. 103, 103706], concluding that this upturn should be related to impurity or boundary scattering. The authors of the current manuscript should comment on these standard metallic behaviors observed in the resistivity curves, and appropriately cite the literature.

iii) In the ARPES study, Ref. [14], it was found sharp quasiparticle peaks and a well-defined Fermi surface characterizes the hole pockets. Moreover, the study reported elliptical electron-like pockets. Given that the marginal Fermi liquid MFL predicts a vanishing quasiparticle weight at the Fermi level, what would be the interpretation of these sharp ARPES features in the context of the MFL phenomenology used in the current manuscript? The authors just say that "we have not yet reached this low- T regime". When this limit should occur and why according to their expectations?

iv) It seems that the coupling to lattice degrees of freedom cannot be so easily excluded and that electron-phonon scattering may provide a significant contribution to the total scattering rate. In general, the phonon subsystem that characterizes SrIrO₃ should consist of many different optical and acoustic phonons. The authors observe many Raman active phonon modes, few of them involving rather small optical phonon energies. Furthermore, it is established in the literature that the bands in SrIrO₃ are quite sensitive on in- and out-of-plane octahedral rotations, while the authors suggest a high sensitivity of flat bands associated with the hole-like Fermi surface to structural details. However, in the manuscript there is no mention of the coupling to phonons and no explanation of reasons for neglecting this coupling when considering the relaxation rates.

v) Measurements of temperature-dependent resistivity would be quite helpful for the manuscript, as

an independent check of the results obtained from the Raman response. This, or other transport measurements, would be particularly beneficial for the interpretation of results presented in Table 1 and Fig. 5.

A revised version of the manuscript should elaborate on these points with the aim of achieving a consistent physical picture of the studied system. This would secure a quality that is appropriate for manuscripts that aim at publication in Nature Communications.

Reviewers' comments:

Reviewer #1 (Remarks to the Author):

This manuscript reports Raman measurements in polarized light on the SrIrO₃ thin films that focus on the electronic Raman response. A careful and methodical analysis of the electronic Raman responses using the memory function formalism originally used by Opel et al in 2000 and developed here, leads the authors to identify two kinds of carriers, hole and electron pockets, depending on the polarizations of light used.

They say that the Raman responses of these carriers cannot be interpreted as a Fermi Liquid but rather as a marginal Fermi liquid. From this point of view, they consider that the charge dynamic in SrIrO₃ is similar to that of strongly correlated systems and they found that the static scattering rate versus T follows the Planckian limit observed from transport data in cuprates. By this study the authors highlight the power of Raman spectroscopy for studying the carrier dynamics and in particular, scattering rate, effective mass of charge carriers.

The authors have carried out a very fine experimental study and an analysis of the Raman data using the formalism of the memory function which opens new perspectives for the exploitation of electron Raman spectroscopy.

However, there are several points which concern me and which do not seem clear enough for me to be in favour of an immediate publication.

The first point is the authors' assertion that Raman response cannot be interpreted in the context of a Fermi liquid model. The authors show curve fits in figures S4 and S5 of the supplementary information without giving the values of the parameters. what are the values of g , D , β , and λ ? To what extent by varying g and /or λ in the FL model, it would not be possible to better fit the low energy part of the Raman spectra in figures S4 and S5 and so attenuate or even make the hump disappear around 100 cm⁻¹?

Our answer: We thank the referee for highlighting this point. The failure of the Fermi liquid (FL) model has indeed been pivotal in developing the analysis presented in the paper. It is important to point out that the Raman response of a weakly interacting electron gas does not display any broad continuum and gives contributions only for frequencies below $q \cdot v_F$, where q is the finite photon momentum. The observation of a broad electronic continuum clearly rules out such a weak coupling description of our data. Alternatively, the interaction with some collective modes can give rise to a momentum independent self-energy and hence to a memory function where the scattering rate depends on the square of the frequency or temperature, see Eq.5 in main text. Our subsequent fits can also rule out this more general notion of a FL.

Now, and to answer precisely the remark of the referee, we recall that λ is associated with the mass enhancement factor, $m^*/m_b(\omega, T) = 1 + \lambda(\omega, T)$. λ is **not** a fitting parameter, but a function of fitting parameters g , β , D and Γ_{imp} , as stated in the supplementary information (see equations 10 and 12, for the FL and NFL models, respectively). In FL model, β is exactly set to 2π , and D is a fixed cut-off frequency which equals to the bandwidth of 250 meV for the relevant hole-like band (details are given in the supplementary information) so that only g and Γ_{imp} can be used as free parameters to fit the data within the FL model.

The curves in Fig. S4 display representative FL fit to the data. The corresponding values of g (1.8) and Γ_{imp} (14.6 meV) are now specified in the revised caption of that figure. To best illustrate the effects of these two parameters, we show the results obtained by varying them independently in Fig. R1 below. Increasing values of g or decreasing values of Γ_{imp} both yield a sharpening of the low energy hump. On the contrary, decreasing values of g or increasing values of Γ_{imp} push the hump towards higher energies, so that it is never possible to correctly fit the low temperature part of the data.

Figure R1: (a) Raman scattering response in XY at 20 K (symbols). The corresponding global fit (solid line in black) with Fermi liquid (FL) scattering rate. Electronic Raman response (χ_e'') for FL is also shown (solid line in red). Additionally, we simulated χ_e'' for FL with the correlation strengths $g=0.5$ and 5.0 . The other parameters are $\beta=2\pi$ and $\Gamma_{\text{imp}}=14.6$ meV. Clearly, g is unable to suppress the low-energy hump in the FL model. (b) We simulated χ_e'' for FL for various Γ_{imp} , in which the common parameters are $\beta=2\pi$ and $g=1.8$. Conclusively, the low-energy hump is independent of the values of Γ_{imp} .

Note that this analysis is quite tedious due to the presence of phonons for which the fitting parameters also sensitively depend on the exact shape of the electronic background (this is true for the regular mode, and even more for the one with the Fano lineshape). Using a systematic iterative routine for MFL, we can reasonably subtract the phonons and attempt to fit the extracted pure electronic Raman responses (χ_e'') in X'Y' with FL model.

As shown in Figure R2(a) for $T=20$ K, impurity scattering rate up to $\Gamma_{\text{imp}}=50$ meV are unable to destroy the low-energy hump. Subsequently, we allowed g and Γ_{imp} to take any values that can attenuate the low-energy hump and offers a better fit over a wide energy range. The best fit is obtained by a combination of $g=9.55$ and $\Gamma_{\text{imp}}=405$ meV which strongly underestimates the hump to fit better the high-energy part. However, such a large impurity scattering rate is not consistent with the DC transport properties of our sample, and also $g=9.55$ gives rise to an unusually large mass enhancement of $m^*/m_b(\omega \rightarrow 0) = 10$. Furthermore, Γ_{imp} is by definition a

temperature independent quantity, but the value obtained at low temperature cannot yield satisfactory agreement at room temperature.

Figure R2: (a) Electronic Raman response (χ''_e) in XY at 20 K (symbols). The fits (solid lines) are according to the Fermi liquid (FL) model for $\Gamma_{imp}=14.6$ and 50 meV. Additionally, the fit shown in dark line was obtained when g and Γ_{imp} were both set free. The parameter $\beta=2\pi$ for all cases, as it is the property of FL. (b) The corresponding data and fits at $T=310$ K. Here, all Γ_{imp} were fixed to the corresponding values at $T=20$ K.

Reviewer 1: Importantly, if one of the main conclusion of the paper is that the dynamics of the carriers in SrIrO3 thin films cannot be interpreted in terms of FL expression. I expect that at least, the S4 or S5 figure (with the parameters used) to be put in the main text and not in the supplementary information.

Our answer: Following the comment of the referee, we have now added the Figure S4 (with parameters used) in the inset of the revised Figure 3a in the main text (page #22).

Reviewer 1: In figure 3, which show the different stage of the process for extracting the Raman electronic background in X'Y' configuration, is hard to understand why in panel (c) is plotted the temperature dependence of the electronic XY Raman response and not the one of X'Y'? Can the authors show the electronic XY Raman response and if they can't do it, they have to explain why?

Our answer: Following the referee's suggestion, we have thoroughly reorganized Figure 3 and we have shown the global fit in XY configuration.

Reviewer 1: In view of the electronic Raman data, the authors propose that SrIrO3 is a strongly correlated material as it is the case of the cuprates. If this is actually the case, I would like the authors indicate or suggest which kind of strong electronic interactions occur in SrIrO3? SrIrO3 is a non-magnetic material, the correlations likely do not come from antiferromagnetic fluctuations as in cuprates, so, why make the analogy with cuprates? I would like the authors to be clearer on this point.

Our answer: As specified in the manuscript, the electronic correlations are quantified by the dimensionless coupling parameter g (introduced with eq. 6 in main text), which is the ratio of the onsite Coulomb repulsion (U) to the bandwidth (W). Our analysis revealed that g amounts to ~ 1.2 and ~ 0.5 for hole- and electron-pockets in SIO, respectively. In comparison, g for YBa₂Cu₃O₇ and Bi₂Sr₂CaCu₂O₈ are 0.55 and 0.4, respectively. Arguably, U is much weaker in

SIO (and 5d compounds in general) than in the cuprates, but the strong spin orbital coupling (SOC ~ 0.4 eV) narrows down the bandwidth, yielding the large g value. It was further shown that perovskite iridates can find themselves relatively close to a Mott-insulator state even with a modest U , as in presence of strong SOC the required critical U to drive a metal-to-insulator transition is strongly reduced [Watanabe et al. Phys. Rev. Lett. **105**, 216410 (2010) – now added to the references].

Finally, we note that even though SrIrO_3 is a paramagnetic metal, it is not far from an AF instability that can be induced using e.g. confinement [Matsuno et al. PRL **114**, 247209 (2015)]. It would not be surprising to observe AF fluctuations in this system, akin to doped Sr_2IrO_4 , in which AF paramagnons survive the suppression of the AF order. [Gretarsson et al. PRL **117**, 107001 (2016)].

We have clarified that discussion and included the reference: Watanabe et al. Phys. Rev. Lett. **105**, 216410 (2010) in main text (page #9, #10).

Reviewer 1: Finally, I have a couple of questions:

-How the authors know the system they studied is free of constraints between the substrate and the thin film?

Our answer:

In previous work (Kleindienst et al. Phys. Rev. B **98**, 115113 (2018)), some of us have investigated structural and transport properties of SIO films grown on various substrates, for examples (101)-oriented orthorhombic DSO, (001)-oriented cubic STO and LSAT substrates. In particular, we obtained that the lattice parameters of such thick SIO films are comparable. Moreover, their deviations with respect to the ones of bulk SIO is $<1\%$, as listed in Table R1. This already indicates that the substrate-induced epitaxial strain impact on the structural property of such thick films is minimized, except for the fact that SIO films are crystallographically twinned when grown on cubic substrates (STO or LSAT).

	a (Å)	b (Å)	c (Å)	Δa (%)	Δb (%)	Δc (%)
SIO on DSO	5.59	7.92	5.61	+0.35	+0.38	+0.17
SIO on STO	5.58	7.82	5.58	+0.17	-0.88	-0.35
SIO on LSAT	5.60	7.82	5.60	+0.53	-0.88	0
Bulk SIO	5.57	7.89	5.60	-	-	-

Table R1: Structural properties of SIO thin films grown on various substrates. Lattice parameters are assigned in $Pnma$ space group (the Δ give the comparison to bulk polycrystalline samples).

The comparable structural properties in these films are also reflected in their transport properties. For instance, absolute values of resistance (R) and general resistance versus temperature (R - T) characteristics are comparable for the films on STO and DSO substrates, as shown in Figure R3(a). Furthermore, in a recently published article (Jaiswal et al. AIP Advances **9**, 125034 (2019) – now cited in the present paper), some of us have shown that the magneto-

resistances of 9 and 50 nm thick SIO films on (101)-DSO are very similar (see also in Figure R3(b)-(c)), indicating that the electric transport of SIO films is already strain-independent, at least from the thickness of 9 nm onwards. Thus, this strongly manifests that the electric transport property of these SIO films are free from substrate-induced epitaxial strain which most possibly is confined within a few monolayers of SIO next to the substrate surface.

Figure R3: (a) Resistance versus temperature characteristics of ~ 60 nm thick SIO films grown on (101)-oriented DSO and (001)-oriented STO substrates. Magneto-resistance of (b) 50 nm and (c) 9 nm thick SIO films grown on (101)-oriented DSO substrates (Jaiswal et al. *AIP Advances* 9, 125034 (2019)).

Reviewer 1: how the authors eliminate the Raman vibration modes of the air molecules at the second focus point in the confocal set-up?

Our answer: The Raman spectrometer that we used (LabRAM HR from Horiba-Jobin Yvon) has an interesting design in this respect. The mirror before the first confocal point acts as an elastic line filter (notch filter), so that the second focus point is only reached by inelastically scattered photons. Their weak intensity does not produce any parasitic air signal (in contrast with other spectrometers). To highlight this, we have updated Fig. S1 and its caption in supplementary information.

Reviewer 1: Why the phonon peaks are smaller in intensity in the XY configuration with respect to the X'Y' configuration?

Our answer: The two scattering geometries probe modes with different symmetries: XY geometry probes phonon modes of B_{1g} and B_{3g} symmetries in SIO (D_{2h} point group, Pnma space group), whereas, X'Y' geometry probes phonon modes of A_g and B_{2g} symmetries.

Predicting the Raman intensity is a complex task, but this is not unusual in these materials to have symmetries that yield much less intensities. For instance, previous reports for isostructural compounds SrRuO₃ (films), LaMnO₃ and YMnO₃ (crystals) have shown that the inherent intensity of B_{1g} and B_{3g} modes is much weaker than the intensity of A_g and B_{2g} modes [Phys. Rev. B **57**, 2872 (1998), Phys. Rev. B **59**, 364 (1999)]. In particular, B_{1g} and B_{3g} modes could not be detected in SrRuO₃ films.

Reviewer 1: The authors mentioned a double phonon around 600 cm⁻¹ but I don't see any single phonon at 300 cm⁻¹. Can the authors explain why?

Our answer: This assignment is based on the weak temperature dependence and the featureless lineshape of the peak, which is located above the highest calculated frequency of the phonon spectrum for SrIrO₃. The detailed analysis of two-phonon Raman response can be tedious. Strictly speaking and as pointed by e.g. M.V. Klein on p.123 of seminal textbook 'Light Scattering in Solids III' (edited by M. Cardona and G. Güntherodt, Springer-Verlag, Berlin) the selection rule mentioned by the referee is only valid for fully symmetrical A_{1g} modes, as all overtones are required to have components with A_{1g} symmetries. This is not true anymore for lower symmetries such those investigated here. Overall, the two phonon-scattering process is mainly a density-of-states effect which can but does not have to involve zone-center phonons, typically two phonons with opposite momenta from the high-density of states region of the dispersion. This is specified in the revised main text (page #6).

Reviewer #2 (Remarks to the Author):

This paper presents the study of electronic dynamics in the strongly correlated Dirac semimetallic SrIrO₃. Combined with the inelastic light scattering and ab-initio calculation, authors analyzed the electronic excitation in the scheme of the marginal Fermi liquid phenomenology. They show that the spectral shape can be modelled by using the marginal Fermi liquid theory, but the temperature dependence of scattering rate is not fully understood by the existing model. Although there are some issues to be addressed, the results are interesting. In my opinion, this manuscript may deserve the consideration in Nature Communications, after addressing the following issues.

1) SrIrO₃ is a Dirac semimetal which possesses electrons with Dirac-like dispersion and holes with conventional band dispersion. An important issue of this material is the dynamics of Dirac electrons. Authors clarified that the renormalization is slightly smaller for Dirac electrons than for holes. Is there a special mechanism that the renormalization of Dirac electron is smaller than that of holes?

Our answer: we thank the referee for this interesting remark. As both seen in the ARPES experiments and in the first principles calculations, the electrons have a substantially larger Fermi velocity than the holes (which have larger effective mass), which results in a small density of states (DOS). As pointed out in the revised version of the main text (page #9), qualitatively, the smaller DOS of the electrons reduces interaction effects and explain the different behavior.

From our calculation the total DOS for the electron bands amounts to 1.6 states/eV (per unit-cell), against 10.7 for the holes. The ratio between the coupling constants for holes ($g \sim 1.2$) and electrons ($g \sim 0.5$) from our analysis is $1.2/0.5 \sim 2.4$, is of the same order of magnitude than that of the DOS ($10.7/1.6 \sim 6.7$). To go beyond this simple qualitative argument, a more elaborate analysis involving explicit interaction matrix elements and/or the treatment of interband scattering events (that would always tend to lower the distinction between the two bands) should be performed, but goes well beyond the scope of the present paper.

Reviewer 2: 2) Authors present that the temperature dependence of scattering time appears to be proportional to T in all the temperature region and does not show the “low- T regime of marginal Fermi liquid theory”. This is an interesting result, but the reason is not clear. Why is “the low- T regime” missing in this material? The materials nearby the quantum critical point often show such T -linear behavior. Is SrIrO₃ is related to the quantum criticality?

Our answer: This is indeed an interesting issue. At the heart of it is the qualitatively different frequency and T -dependence of the scattering rate Γ (believed to be proportional to the resistivity) and the quasi-particle scattering rate $\tau^{-1} = \Gamma/(1 + \lambda)$ as introduced in Eq. (1). This distinction is due to the T -dependence of the effective mass, encoded in λ , and occurs for $\alpha \leq 1/2$, including the MFL case $\alpha = 1/2$. While the resistivity of a MFL would be linear in T , the quasiparticle scattering rate behaves like $\tau^{-1} \propto T/\log(\frac{D}{T})$, caused by the well-known logarithmic divergence of its effective mass. Planckian behavior is judged based upon the quasi-particle scattering rate τ^{-1} . This implies that, strictly speaking, a MFL does not show pure Planckian behavior. The latter would only occur for $\alpha < 1/2$, where $\tau^{-1} \propto T$ independent of α . In the Figures below we show the T -dependence of λ , Γ , and τ^{-1} at zero frequency as deduced from the fits to our finite-frequency data.

Figure R4 Temperature dependence of the effective mass enhancement λ , scattering rate Γ , and quasiparticle scattering rate τ^{-1} as deduced from our fit to Raman spectra. While the slope of the scattering rate is rather different for electrons and holes, they become more comparable for the quasiparticle scattering rates and comparable to the Planckian time. Notice, τ^{-1} of the MFL is not strictly linear but affected by the logarithmic divergency of the mass enhancement.

While the slope of the scattering rate is rather different for electrons and holes, they become more comparable for the quasiparticle scattering rates and comparable to the Planckian time. Notice, however, that τ^{-1} of the MFL is not strictly linear but affected by the logarithmic divergency of the mass enhancement. The increase of the effective mass as function of frequency, shown in Fig.4, of the manuscript is however fully consistent with MFL behavior.

We have rephrased the corresponding sentence which now reads:

At the lowest temperatures, the logarithmic growth of the MFL mass renormalization yields $\hbar \tau^{-1} \sim \frac{\pi \beta}{2} \frac{k_B T}{\log \frac{D}{\beta T}}$, where holes and electrons reach different quasiparticle rates. In the temperatures range investigated here, however, the MFL mass renormalization remains limited $\lambda(T) < 3$ (see supplementary information), and the log divergence of $\hbar \tau^{-1}$ is not detectable.

The relationship to quantum criticality is a possibility (related e.g. to nearby magnetism). It could be one of the ‘underlying ordering principles’ we mention. In the absence of evidence for this (to the best of our knowledge), we have however chosen to not explicitly discuss this.

Reviewer 2: 3) The ARPES study reveals that there is a hole pockets nearby the Gamma-point in the momentum space, which substantially contributes the DC transport. In this work, authors do not mention the hole pocket nearby Gamma-point since the Raman scattering probes the pockets at the zone boundary. Nevertheless, in the final part of the main text and Figure 4, authors compare the results of Raman scattering with that of DC-transport. The missing information about hole pocket at Gamma-point may be critical for this comparison. It would be better to append the explanation about the comparison.

Our answer: The ARPES study in Nie et al. Phys. Rev. Lett. **114**, 016401 (2015) indeed revealed that there is a hole-pocket at the Γ -point of the 1-Ir Brillouin zone (BZ) for pseudocubic unit cell. We also captured that pocket in the 1-Ir BZ from our DFT calculations (see Figure 1b in main text). Note that the hole-pocket at the Gamma-point is basically the same pocket that repeats at $(0, \pm\pi)$ in 1-Ir BZ. Therefore, we determined dynamic scattering rate and mass enhancement factor for all the hole-pockets at the zone boundaries and at the Γ -point of 1-Ir BZ. In order to avoid similar confusion that readers may have, we added this information in the caption of Figure 1b in the revised main text (page #20).

Reviewer 2: 4) Authors suggest that the electronic structure at the Fermi surface does not significantly change as a function of temperature. However, this statement may not be consistent with the previous research by optical conductivity which reveals a substantial change of electronic structure as a function of temperature. It would be better to append some explanation.

Our answer: We thank the referee for this interesting remark. Indeed, optical conductivity measurements on polycrystalline SrIrO₃ from [Phys. Rev. B **95**, 121102R (2017)] show a massive spectral-weight transfer around 100 meV, which is interpreted as ‘a significant reconstruction of the electronic structure with temperature mainly occurs in the occupied $j_{eff} = 1/2$ state and the $j_{eff} = 3/2$ state located near E_F , while keeping intact the unoccupied $j_{eff} = 1/2$ state and the $j_{eff} = 3/2$ state located far from E_F ’.

This calls out for a couple of comments. On the one hand, the temperature dependent spectral-weight transfer seen in optical conductivity is associated with an interband transition to which electronic Raman scattering might be insensitive (because the selection rules for interband optical absorption and interband Raman scattering are complimentary in centrosymmetric systems like SIO: interband optical absorption involves bands with opposite parity, whereas interband Raman scattering requires the bands with same parity, see e.g. Phys. Rev. B **4**, 2429 (1971)). But even in this case, one would expect a reconstruction of the electron pockets to be reflected in the intraband charge dynamics we are measuring. This could indicate that, contrary to the claim of [Phys. Rev. B **95**, 121102R (2017)], the reconstruction is occurring away from E_F .

On the other hand, this electronic reconstruction in the polycrystalline samples of this report is also associated with enhanced paramagnetism and magnetoresistivity, two effects that are absent in our thin films. One key difference pointed out in this paper is that the Dirac node in the polycrystalline samples is about twice closer to E_F than what is reported for the films. In a recent work [Appl. Phys. Express **13**, 015510 (2020)], we became aware of in the process of replying this reports, optical conductivity measurements on metallic thin films grown on STO display a very reduced spectral weight transfer compared to the polycrystalline samples (see fig. below – the relative change of the total spectral weight is much lower in the films). This can clearly be attributed to subtle structural differences.

[Redacted]

Figure R5 Optical conductivity data of polycrystalline (Phys. Rev. B 95, 121102R (2017)) and STO-grown thin film (Appl. Phys. Express 13, 015510 (2020)) SrIrO₃ sample. The plot have been rescaled for direct comparison and show that the relative transfer of spectral weight is strongly reduced in film in comparison with the polycrystalline material.

As pointed it in the reply to referee#1 and #3, even though strain effects are minimized in the films by the choice of DSO substrate, structural details of the unit cell are slightly different between the films and the polycrystalline samples (see Table R1), which in turn affects the electronic structure and the possible instabilities it might undergo.

Thus, a more likely explanation of the difference between the conductivity and Raman results is the nature of the investigated samples.

We have added a comment to this point in the revised version of the manuscript (page #9).

Reviewer #3 (Remarks to the Author):

The manuscript, entitled “Strange semimetal dynamics in SrIrO₃”, brings a Raman study of SrIrO₃, a member of iridate perovskites from the Ruddlesden-Popper series. Unlike other members of that series, which are layered, SrIrO₃ is a three-dimensional system, exhibiting (semi)metallic behavior.

The two most important achievements presented in this manuscript are:

- i) Independent measurements of the electron- and hole-like contributions to the Raman response, by probing different sections of the FS.
- ii) A fitting procedure that provides a relatively convincing case for a phenomenological description of the charge dynamics in terms of a scattering rate that is linear in frequency and temperature.

I think that these results are interesting and eventually worth publishing in Nature Communications. However, I have some issues with the consistency of the new findings with the already published results. Furthermore, I have worries regarding the physical interpretation of the experimental data. In the light of these concerns, I would like to draw attention to few essential problems, mostly in relation to the nature of the scattering rate. Specifically:

- i) How important are the strain-relaxation effects for the present study? In the part of the manuscript that discusses experimental aspects it is said that fully strain-relaxed 50 nm thick films are measured, suggesting that the electron band structure and the Raman response are not affected by the film thickness. Are there any data to confirm this?

Our answer: As pointed out in our introduction, strain effects can be critical in iridates and yield controversial results. We have carefully chosen the substrate and the thickness of the film to ensure sure they were not important for the present study.

To limit strain effects, DSO is arguably the best substrate. Indeed, the lattice mismatch between the (101)-DyScO₃ (DSO) substrate and bulk-SIO along the in-plane directions of DSO are only 0.25 % (tensile along [0 1 0]) and 0.06 % (compressive along [-101]). X-ray diffraction (XRD) results yield that the lattice parameters of the representative SIO films are close (deviation: <1 %) to the ones found in bulk-SIO, as listed in Table R1. In addition, the tilt pattern of the IrO₆ octahedra, $a^+b^+a^-$ in Glazer notation, is the same as that of bulk-SIO (detailed in Ref. 21).

Secondly, some of us have shown in recently published work [AIP Advances **9**, 125034 (2019)] that the DSO substrate-induced strain effects, if any, are mostly confined within a few monolayers of SIO. There magneto-resistances of 9 and 50 nm thick SIO films on (101)-DSO are very similar (see them in Figure R3(b)-(c) in this report), indicating that strain effects for thicknesses larger than a few nm are negligible. This was confirmed by the comparison of the transport properties of ~60 nm thick SIO films grown on (101)-DSO and (001)-STO. Even though the two substrates impose slightly different strain to the film, the results are very similar (see in Figure R3(a)).

Raman scattering on films grown on STO substrate are challenging because of the strong STO contribution to the signal. Along this line, we also measured 50 nm thick SIO films grown on (001)-LSAT substrate. We obtained that the overall Raman scattering response of these films can be compared well with the reported Raman response of the SIO films grown on (101)-DSO substrate (see Figure R4). The only difference is that the SIO films grown on these cubic STO or LSAT substrates are twinned, i.e. the crystallographic *b*-axis of SIO is oriented along both

in-plane directions ([100] and [010]) of these cubic substrates, and of lesser crystalline quality. This results in weaker Raman intensity and potentially complicates the interpretation of scattering data.

Figure R6: Raman scattering response of 50 nm thick SIO films on (101)-DSO and (001)-LSAT substrates in X'Y' polarization configuration at 20 K.

Reviewer 3: ii) It appears that there is a good agreement in the literature that bulk SrIrO₃ exhibits a metallic resistivity [PRB 89, 214106; PRB 95, 121102(R); J. Appl. Phys. 103, 103706]. In particular, in the recent study [PRB 89, 214106], for low temperatures ($T < 30\text{K}$), it was found that the bulk resistivity follows the typical Fermi-liquid behavior quadratic in temperature. Above these temperatures the resistivity became linear in T , which is typical for metals and for the scattering on thermal phonons. The authors of [PRB 89, 214106] find no indication for an upturn in resistivity below 50 K reported in [J. Appl. Phys. 103, 103706], concluding that this upturn should be related to impurity or boundary scattering. The authors of the current manuscript should comment on these standard metallic behaviors observed in the resistivity curves, and appropriately cite the literature.

Our answer: We thank the referee for this remark. It is correct that a metallic resistivity has been reported in bulk (but polycrystalline) materials, and that the shallow minimum or upturn in ρ observed at low temperature in e.g. [Phys. Rev. B 95, 121102(R) (2017); J. Appl. Phys. 103, 103706 (2008)], is characteristic of Anderson type charge carrier localization in presence of impurity and boundary scattering, crystal defects due to Ir- and/or O-vacancies, distortions etc... As argued in our reply to the previous point, even though we pay special attention to the choice of the substrate to limit strain effects, the lattice parameters of the single crystalline phase stabilized in thin film form remain different than that of the polycrystalline material. The latter have indeed a smaller unit cell (the pseudocubic unit cell volume of the film of this study are 0.65% larger than that of e.g. Phys. Rev. B 89, 214106), and it is therefore not surprising to

observe different behavior in films and polycrystalline materials. This provides another example of the extreme sensitivity of the electronic properties of SrIrO₃ to structural effects, which is rooted in the (semimetallic) narrow band-structure of this material close to a metal-to-insulator transition.

To account for the reviewer remark, we have incorporated the missing references and a mention to the polycrystalline materials in the introduction of the revised manuscript (page #3).

Reviewer 3: iii) In the ARPES study, Ref. [14], it was found sharp quasiparticle peaks and a well-defined Fermi surface characterizes the hole pockets. Moreover, the study reported elliptical electron-like pockets. Given that the marginal Fermi liquid MFL predicts a vanishing quasiparticle weight at the Fermi level, what would be the interpretation of these sharp ARPES features in the context of the MFL phenomenology used in the current manuscript? The authors just say that “we have not yet reached this low-T regime”. When this limit should occur and why according to their expectations?

Our answer: This is an interesting question. Clearly, at T=0, the quasiparticle weight of a MFL state vanishes and the one-to-one mapping between electronic states of the many-body system and of a free electron gas, which is the foundation of Landau’s theory of Fermi Liquid, breaks down. However, this does not imply that an ARPES measurement could not observe well defined peaks in the spectral function. To demonstrate this behavior, we plot the spectral function of a Fermi liquid and a Marginal Fermi liquid at a small but finite temperature for different momenta as the bare band crosses the Fermi energy. We used the same parameters for the coupling constant, T and upper cut off D in both cases.

While the MFL is clearly broader than the FL, one can still identify well-defined peaks that sharpen as one crosses the Fermi energy at $\omega=0$. Hence, seeing a well-defined peak in ARPES does, by itself, not imply that the system is a FL or rule out MFL behavior. To draw such a conclusion would require a detailed analysis of temperature and energy dependence of the line shape of the quasiparticle peak. We agree with the referee that our findings clearly call for a more refined analysis of ARPES data in SrIrO₃ and hope that our work will stimulate such investigation.

Figure R7: calculated spectral function $A(\omega)$ at finite temperature using the same e-e coupling constant and upper cut-off for a Fermi Liquid and a Marginal Fermi Liquid. Both show clear quasiparticle peaks.

Reviewer 3: iv) It seems that the coupling to lattice degrees of freedom cannot be so easily excluded and that electron-phonon scattering may provide a significant contribution to the total scattering rate. In general, the phonon subsystem that characterizes SrIrO₃ should consist of many different optical and acoustic phonons. The authors observe many Raman active phonon modes, few of them involving rather small optical phonon energies. Furthermore, it is

established in the literature that the bands in SrIrO₃ are quite sensitive on in- and out-of-plane octahedral rotations, while the authors suggest a high sensitivity of flat bands associated with the hole-like Fermi surface to structural details. However, in the manuscript there is no mention of the coupling to phonons and no explanation of reasons for neglecting this coupling when considering the relaxation rates.

Our answer: we thank the referee for highlighting this interesting point. We do indeed observe many Raman active optical phonons that mostly behave normally (and will be the subject of a separate publication), and as explained in the manuscript, only two of these modes display a clear Fano lineshape at fairly high energy 400 cm⁻¹ or 50 meV, indicating their coupling to the electronic background. It is therefore legitimate to wonder whether the scattering of the electron on these phonons does not contribute to the total (static) scattering rate, akin to what is commonly seen in *e.g.* resistivity.

It is true that on general grounds, there is indeed no reason to exclude such contribution. For temperatures much lower than the Debye temperature, which is determined by the bandwidth of the phonon dispersion (which amounts to ~70 meV in SrIrO₃ so 800K), electron-phonon coupling is expected to contribute to the total static scattering rate in two ways (see e.g. J. Bass, *et al.* Rev. Mod. Phys. 62, 645 (1990)): through a ‘normal’ Bloch T^5 term and through a more complex umklapp scattering term $\Gamma_u \propto e^{-\hbar\Omega/k_B T}$ (where Ω is the frequency of the phonon with minimum wave vector that allows electrons to scatter through an umklapp process - typically few meV).

Now, even looking at the raw data, we can discard these processes as a sizeable contribution to the scattering rate. Indeed, the slope of $\chi''(\omega)$ as $\omega \rightarrow 0$ is directly proportional to the inverse of the scattering rate. Clearly the scattering rate obtained this way (see Fig. R8 for the case of the hole carriers) already has an almost linear dependence with temperature, and this trend can simply not be accounted for by the aforementioned electron-phonon scattering terms.

Figure R8: low energy slope of the Raman response in X'Y' channel (hole carriers)

Adding the e-ph terms to a Fermi Liquid like analysis (see reply to referee 1) yields an overall increase of the scattering rate at low frequencies, which is already too large to reproduce the data. The presence of the continuum extending up to very high frequencies makes it absolutely necessary to phenomenologically modify the electron-electron scattering rate function and we were able to show that the MFL chosen form yields already a very good agreement with the data. Adding complexity

to the analysis through additional e-ph terms in the memory function analysis does therefore not appear justified.

Reviewer 3: v) Measurements of temperature-dependent resistivity would be quite helpful for the manuscript, as an independent check of the results obtained from the Raman response. This, or other transport measurements, would be particularly beneficial for the interpretation of results presented in Table 1 and Fig. 5.

Our answer: The referee may have overlooked the temperature-dependent resistivity of the present sample. We had originally shown this in Figure S9 and discussed this in the main text.

Reviewer 3: A revised version of the manuscript should elaborate on these points with the aim of achieving a consistent physical picture of the studied system. This would secure a quality that is appropriate for manuscripts that aim at publication in Nature Communications.

Our answer: We thank the reviewer once more and hope he/she will find the revised version of the paper appropriate for publication in Nature Communications.

REVIEWER COMMENTS

Reviewer #1 (Remarks to the Author):

Dear editor,

The authors performed improvements to their manuscript. In addition, the authors have answered all my questions and taken my suggestions into account. I also noticed that all the points raised by the other referees have been treated fairly and seriously by the authors. In my opinion, the manuscript is now ready for publication.

Reviewer #2 (Remarks to the Author):

The authors have improved the paper according to my comment and suggestions. Most of issues are properly addressed and the manuscript can be accepted for publication.

Reviewer #3 (Remarks to the Author):

The authors have provided a detailed response to my questions raised in the first report, making their case stronger. Some parts of the manuscript have been improved still focusing of high quality Raman measurements of SrIrO₃ films and the phenomenological interpretation of the experimental data.

It seems to me that the weakest point of the manuscript is the lack of experimental confirmation of the low-temperature MFL regime. As clearly stated in the new version of the manuscript, "the MFL mass renormalization remains limited, and the log divergence of is not detectable". Therefore, I am repeating my question about the energy scale D which might govern this MFL limit. What is the limiting D to finite values, since the MFL theory is introduced purely phenomenologically? Without any knowledge of D , the extensive usage of the MFL term, all over the manuscript, might be misleading for readers. It would be better to use terms like "MFL-like behavior" or "linear T dependency", etc.

In the new version of the manuscript, the authors explain that "subtle structural differences between the polycrystalline materials and the thin films can yield different behavior at low temperatures (electronic structure reconstruction, location of the Dirac node, metal-to-insulator transition, sign of the magnetoresistance)", and, in particular, that "slight changes in the lattice parameters can induce metal-to-insulator transition". In other words, behavior of SrIrO₃ systems may be quite dependent on details of the preparation and hence conductive properties may differ from a study to a study. This puts limitations on the generality of the results, which should be more clearly pointed out in the final version of the manuscript.

Since the SrIrO₃ systems are so sensitive to structural differences, I would not so easily exclude the possibility that lattice degrees of freedom play a role, as suggested by authors in their response. In the reference [Rev. Mod. Phys. 62 645], cited by the authors in their response to my comments, one finds that the 'normal' Bloch term involves "a linear variation of the resistivity with T , from the melting point down to below the Debye temperature Θ , and an ultimate T^5 variation for $T < \Theta/50$ ". That is, the T^5 behavior of the resistivity due to the scattering on phonons is limited to the very low temperatures, whereas the elevated temperatures are characterized by a linear dependence in T . Taking the value of Θ quoted by the authors, $\Theta = 800\text{K}$, the T^5 behavior is observable below 20K, which is irrelevant for the current study since it does not consider this low-temperature range at

all.

The Bloch-Gruneisen model [Rev. Mod. Phys. 62 645] has been derived for the scattering on acoustic phonons in a standard metal, assuming a single band and a large Fermi energy. If we listen to the authors, the system under investigation involves substantial electron correlations that certainly affect the electron-phonon interaction and, consequently, the scattering on phonons. Thus, the applicability of the simple Bloch-Gruneisen model in the case of SrIrO₃ films is questionable. In order to discuss the origin of the scattering mechanisms, it might be better to consider the interplay between electron-electron and electron-phonon correlations. In any case, the importance of the scattering on acoustic phonons cannot be fully excluded, in addition to effects related to electron quasiparticle properties near the Fermi level. However, I can agree with the authors that "adding complexity to the analysis through additional e-ph terms in the memory function analysis" would not necessarily provide deeper understanding. I am just emphasizing that the origin of the scattering rate is not fully apparent from the data and that different scenarios can be put forward.

In closing, the quality of the experimental research is apparently high. The SrIrO₃ system is interesting and the current study should motivate further investigations that might give us answers on remaining open questions. However, the manuscript would strongly benefit from a more balanced discussion of the phenomenological results. The inclusion of the latter would make this work acceptable for publication in the Nature Communication.

Referee3: The authors have provided a detailed response to my questions raised in the first report, making their case stronger. Some parts of the manuscript have been improved still focusing of high quality Raman measurements of SrIrO₃ films and the phenomenological interpretation of the experimental data.

It seems to me that the weakest point of the manuscript is the lack of experimental confirmation of the low-temperature MFL regime. As clearly stated in the new version of the manuscript, “the MFL mass renormalization remains limited, and the log divergence of is not detectable”. Therefore, I am repeating my question about the energy scale D which might govern this MFL limit. What is the limiting D to finite values, since the MFL theory is introduced purely phenomenologically? Without any knowledge of D , the extensive usage of the MFL term, all over the manuscript, might be misleading for readers. It would be better to use terms like “MFL-like behavior“ or “linear T dependency”, etc.

Our reply: D is set to the bandwidth of each charge carriers because it is the only relevant energy scale of the problem. We have rewritten the sentence of the manuscript to clarify this further. It now reads:

We have set $\hbar D$ to the bandwidth for each charge carrier. In both cases its value amounts to several hundreds of meV, and thereby only weakly affects $\Gamma(\omega, T)$ in the investigated range of frequencies (see Supplementary Note 2).

We should also stress that we do observe strong evidence for MFL behavior, both at frequencies below the energy scale D , including the expected low- ω mass enhancement, and at low temperatures in form of a linear scattering rate. Merely the logarithmic low-T enhancement of the mass, that enters the quasiparticle scattering rate, is beyond the accuracy of our approach. Hence, in our view there is very compelling evidence for MFL behavior. Notice also that the mentioned logarithmic in temperature behavior of the mass enhancement has, to our knowledge, never been observed in the cuprates - arguably very strong candidates for MFL-behavior - either.

Referee3: In the new version of the manuscript, the authors explain that “subtle structural differences between the polycrystalline materials and the thin films can yield different behavior at low temperatures (electronic structure reconstruction, location of the Dirac node, metal-to-insulator transition, sign of the magnetoresistance)”, and, in particular, that “slight changes in the lattice parameters can induce metal-to-insulator transition”. In other words, behavior of SrIrO₃ systems may be quite dependent on details of the preparation and hence conductive properties may differ from a study to a study. This puts limitations on the generality of the results, which should be more clearly pointed out in the final version of the manuscript.

Our reply: The referee is correct that these details matter and that the electronic properties of this class of materials depends very sensitively on minute changes of the lattice structure. This statement is in fact true for many correlated materials and is, in our opinion, largely responsible for their fascinating physics.

It is with this in mind that we have carried out this work on a sample in which strain effects are minimized (after a systematic structural study previously reported in ref. 21: Kleindienst et al. Physical Review B 98, 115113 (2018)). We believe that further systematic investigations of the impact of structural degrees of freedom on the charge dynamics of this interesting system, as it is for instance driven through a metal-to-insulator transition, would be most instructive in assessing the generality of our results. We have added a statement in this respect in the conclusion of the paper, which now reads:

*‘To summarize, in this work we combined the selection rules of polarized Raman scattering and the high spatial resolution of confocal geometry to investigate independently the dynamics of electron- and hole-like charge carriers in a **strain-relaxed** film of semimetallic SrIrO₃ as thin as 50 nm. We find that neither of them can be described within the framework of the Fermi liquid theory, and that the electronic Raman response can be well modelled using marginal Fermi liquid phenomenology. Using mass enhancement and the DC scattering rate obtained from this analysis allow us to retrieve the mobility for the two types of charge carriers. The results confirmed the much larger mobility of the electron carriers that generally dominate transport experiments. **A next natural step would be to investigate how structural degrees of freedom impact this charge dynamics, in particular as SrIrO₃ is driven across metal-to-insulator and/or magnetic transitions using for instance lattice strain tuning.** The proposed approach **more generally** demonstrates the power of Raman scattering to resolve the dynamics of charge carriers in correlated semimetals and more generally multiband systems.’*

Referee3: Since the SrIrO₃ systems are so sensitive to structural differences, I would not so easily exclude the possibility that lattice degrees of freedom play a role, as suggested by authors in their response. In the reference [Rev. Mod. Phys. 62 645], cited by the authors in their response to my comments, one finds that the ‘normal’ Bloch term involves “a linear variation of the resistivity with T, from the melting point down to below the Debye temperature Θ_D , and an ultimate T^5 variation for $T < \Theta_D/50$ ”. That is, the T^5 behavior of the resistivity due to the scattering on phonons is limited to the very low temperatures, whereas the elevated temperatures are characterized by a linear dependence in T. Taking the value of Θ_D quoted by the authors, $\Theta_D = 800\text{K}$, the T^5 behavior is observable below 20K, which is irrelevant for the current study since it does not consider this low-temperature range at all. The Bloch-Grüneisen model [Rev. Mod. Phys. 62 645] has been derived for the scattering on acoustic phonons in a standard metal, assuming a single band and a large Fermi energy. If we listen to the authors, the system under investigation involves substantial electron correlations that certainly affect the electron-phonon interaction and, consequently, the scattering on phonons. Thus, the applicability of the simple Bloch-Grüneisen model in the case of SrIrO₃ films is questionable. In order to discuss the origin of the scattering mechanisms, it might be better to consider the interplay between electron-electron and electron-phonon correlations. In any case, the importance of the scattering on acoustic phonons cannot be fully excluded, in addition to effects related to electron quasiparticle properties near the Fermi level. However, I can agree with the authors that “adding complexity to the analysis through additional e-ph terms in the memory function analysis” would not necessarily provide deeper understanding. I am just emphasizing that the origin of the scattering rate is not fully apparent from the data and that different scenarios can be put forward.

Our reply: We thank the referee for pointing this out. We note however that while the formula for the Bloch-Grüneisen (BG) law given in the cited paper is correct (as recently rederived by one of us for a recent publication – see formula A12 in Levchenko and Schmalian, Annals of Physics 419, 168218 (2020) / arxiv.org/2005.09694), the value ‘ $\Theta_D/50$ ’ from quoted statement is not. As shown in the plot below, the cross-over between the T^5 and the linear T behavior in the resistivity occurs instead around $0,1 \Theta_D$. In other words, should electron-phonon scattering be relevant here (and provided that the BG model is relevant as discussed in the next point), it would be reflected as a change of slope in the scattering rate at intermediate temperatures.

Figure 11: Bloch-Grüneisen scattering. Left panel: Temperature dependence of the Bloch-Grüneisen scattering rate and hence of the resistivity due to electron-phonon scattering. Right panel: scattering rate divided by the leading low- T behavior $\tau_{ep}^{-1} \propto T^5$, demonstrating that it is valid for $T < \theta_D/10$.

So, as much as we generally agree with the referee that there are a priori no reasons to neglect the contribution of the electron-phonon interaction to the scattering, we show in a new supplementary information note that it does not play a dominant role. We are grateful to the referee for stressing this important issue.

To demonstrate that the above statement is correct, we took the inverse of the slope of the electronic Raman response for all the data shown in Fig. 2a and plot it as function of temperature (we normalize this to the room temperature values). As shown in our previous reply, the temperature dependence is roughly linear. If we now compare it with the prediction of the BG law, placing ourselves in the limit in which the scattering at room temperature is entirely caused by acoustical phonon (on top of the temperature independent residual scattering from e.g. lattice defects), we clearly see that over a wide range of temperatures, phonon scattering cannot account for the observed behavior (so in the more realistic case where it would only account for a fraction of the scattering at room temperature, it would generally be negligible).

Of course, it is well known that in conventional metals, the T^5 law is sometimes not observed in transport experiment as phonon drag effects give rise to exponential temperature dependence at low temperature. Those are however caused by momentum conservation that governs the electrical current and are in principle not relevant for Raman scattering.

We are left with the issue that treating electron-phonon scattering in a correlated system is in itself a very complex and challenging issue, that goes well beyond the scope of the present study. Nevertheless, scattering from acoustical phonon will also be governed by phase space considerations and will remain weak at low temperatures.

Given that this scattering mechanism can never account for the presence of the broad electronic continuum that we observed, and which can be described satisfactorily using a marginal Fermi liquid ansatz, we have accumulated significant evidence that it can be reasonably neglected without affecting the conclusions of our study.

To account for the referee's remark, we have added the following statement following introduction of the memory function formalism (p.7):

In the memory function formalism, the experimentally challenging determination of the ERS intensity in absolute units is not required and has not been attempted here. Note that in the following, only electron-electron scattering contribution to the total scattering rate (in particular in the static limit) are included, and possible contributions of long wavelength acoustical phonons to the total scattering rate have not been considered (see supplementary note 5 for a detailed discussion).

We have consequently added a supplementary note 5 in which we discuss this issue in more details.

In closing, the quality of the experimental research is apparently high. The SrIrO₃ system is interesting and the current study should motivate further investigations that might give us answers on remaining open questions. However, the manuscript would strongly benefit from a more balanced discussion of the phenomenological results. The inclusion of the latter would make this work acceptable for publication in the Nature Communication.